# TABGRAPHS: A BENCHMARK AND STRONG BASELINES FOR LEARNING ON GRAPHS WITH TABULAR NODE FEATURES

## ABSTRACT

Tabular machine learning is an important field for industry and science. In this field, table rows are typically treated as independent data samples, but additional information about the relations between these samples is sometimes available and can be used to improve predictive performance. Such information can be naturally modeled with a graph, hence tabular machine learning may benefit from graph machine learning methods. However, graph machine learning models are typically evaluated on datasets with homogeneous, most often text-based node features, which are very different from heterogeneous mixtures of numerical and categorical features present in tabular datasets. Thus, there is a critical difference between the data used in tabular and graph machine learning studies, which does not allow one to understand how successfully graph models can be transferred to tabular data. To bridge this gap, we propose a new benchmark of diverse graphs with heterogeneous tabular node features and realistic prediction tasks. We use this benchmark to evaluate a vast set of models, including simple methods previously overlooked in the literature. Our experiments show that graph neural networks indeed can often bring gains in predictive performance for tabular data, but standard tabular models can also be adapted to work with graph data by using simple graph-based feature augmentation, which sometimes enables them to compete with and even outperform graph neural models. Based on our empirical study, we provide insights for researchers and practitioners in both tabular and graph machine learning fields.

## 1 INTRODUCTION

Tabular data is ubiquitous in industry and science, so machine learning methods for working with such data are of great importance. A key distinction of tabular data is that it typically comprises a mixture of numerical and categorical features that widely vary in their distribution and have different meanings and levels of importance for the task. Such features are called *heterogeneous* or *tabular*. Deep learning methods do not always perform well on datasets with heterogeneous features, so the machine learning methods of choice for tabular data are typically ensembles of decision trees, in particular Gradient-Boosted Decision Trees (GBDTs) (Friedman, 2001). However, there is a growing number of recent works trying to adapt deep learning methods to tabular data (Arik & Pfister, 2019; Badirli et al., 2020; Huang et al., 2020; Gorishniy et al., 2021; 2022; Hollmann et al., 2022).

In tabular machine learning, table rows are usually treated as independent data samples. However, there is often additional information available about the relations between samples, and leveraging this information has the potential to improve predictive performance. Such relational information can be represented as a graph. There are many areas where graphs naturally arise. For example, all sorts of user interactions can be modeled as graphs: social networks, chat applications, discussion forums, question-answering websites, financial transaction networks, etc. And even without direct interactions, meaningful relations between samples can often be defined: for example, connecting users who buy similar products on shopping websites, watch similar content on video hosting services, or perform the same tasks on crowdsourcing platforms. In all these and many other cases, using graph information can improve the quality of predictions made by machine learning models.

Graph machine learning is a field focused on the development of methods for learning on graph-structured data. In recent years, the most successful models for such data have become Graph Neural Networks (GNNs) (Kipf & Welling, 2016; Gilmer et al., 2017). Therefore, it can be desirable to adapt these models to tabular data with relational information. However, GNNs are typically evaluated on graphs with homogeneous features (most frequently, bags of words or text embeddings), which are very different from heterogeneous features present in tabular data. Because of this, it is unclear how successfully these models can be transferred to tabular data. Recently, two methods have been proposed specifically for learning on graphs with heterogeneous tabular node features (Ivanov & Prokhorenkova, 2021; Chen et al., 2022). However, their evaluation setting is limited, since, as the first of these works notes, there is currently a lack of publicly available graph datasets with heterogeneous node features. This highlights the difference between industry, where data with heterogeneous tabular features is abundant, and graph machine learning benchmarks, where such data is barely present. We believe that this difference holds back the adoption of graph machine learning methods to tabular data.

In our work, we aim to bridge this gap. First, we create a benchmark of graphs with heterogeneous tabular node features — TabGraphs. For this benchmark, we collect tabular datasets and augment each dataset with a natural graph structure based on external (i.e., not present in the sample features) information about the data: interactions between users, similar behavior of users, traffic between websites, connections between roads, frequent co-purchasing of products, etc. Our benchmark has realistic prediction tasks and is diverse in data domains, relation types, graph sizes, graph structural properties, and feature distributions. Further, we use this benchmark to evaluate a comprehensive set of machine learning methods. Specifically, we consider models for tabular data (both GBDT and deep learning ones), their versions augmented with graph information, several GNN architectures, their versions augmented with a special numerical feature embedding technique for neural networks, and the recently proposed specialized methods for graphs with heterogeneous node features. Thus, we experiment not only with models used in previous studies but also with several simple methods overlooked in the existing literature. Our main findings are:

- Using relational information in data and applying graph machine learning methods can indeed lead to an increase in predictive performance for many real-world tabular datasets;
- Augmenting the inputs of standard tabular machine learning models with additional graph information is a simple method that often leads to significant performance gains;
- Standard GNNs can provide even stronger results in some cases, but the performance of different graph neural architectures may vary across datasets;
- Augmenting standard GNNs with numerical feature embeddings, which have been proposed in tabular DL as an additional feature processing step, can further improve their performance;
- Standard GNNs and tabular models augmented with graph information outperform recently proposed methods designed specifically for graphs with heterogeneous tabular node features.

We believe that our benchmark will serve two main purposes. First, as discussed above, there are many real-world applications where graph structure can be naturally added to tabular data, and those interested in such applications can test their models on our benchmark. Hence, we hope that the proposed benchmark and new insights obtained using it will lead to a wider adoption of graph machine learning methods to tabular data in industry and science. Second, our benchmark provides an alternative testbed for evaluating GNN performance and, compared to standard graph benchmarks, offers datasets with very different feature types and prediction tasks, which are more realistic and meaningful. Thus, we expect that our benchmark will be useful for researchers and practitioners in both tabular ML and graph ML fields.

Our benchmark and code for reproducing our experiments can be found in our anonymous repository.

## 2 RELATED WORK

**Machine learning for tabular data**    The key distinction of tabular data is that it typically consists of a mixture of numerical and categorical features with vastly different meaning and importance for the task. Standard deep learning models often do not perform well on such heterogeneous features. Thus, the preferred methods for tabular data often involve ensembles of decision trees, such as Gradient

Boosted Decision Trees (GBDTs) (Friedman, 2001), with the most popular implementations being XGBoost (Chen & Guestrin, 2016), LightGBM (Ke et al., 2017), and CatBoost (Prokhorenkova et al., 2018). However, deep learning models have several advantages compared to tree-based ones, such as modularity, ease of integration of different data modalities, the ability to learn meaningful data representations, and the ability to leverage pretraining on unlabeled data. Because of this, there has been an increasing number of works trying to adapt deep learning to tabular data (Klambauer et al., 2017; Arik & Pfister, 2019; Song et al., 2019; Popov et al., 2019; Badirli et al., 2020; Hazimeh et al., 2020; Huang et al., 2020; Gorishniy et al., 2021; Kadra et al., 2021; Müller et al., 2021; Gorishniy et al., 2022; Hollmann et al., 2022; Chen et al., 2023a; Feuer et al., 2024). Further, there is recent research comparing different kinds of tabular models and trying to determine which ones are the best (Shwartz-Ziv & Armon, 2022; Grinsztajn et al., 2022; McElfresh et al., 2023).

Among the tabular deep learning literature, the retrieval-augmented deep learning models (Kossen et al., 2021; Qin et al., 2021; Somepalli et al., 2021; Du et al., 2022; Gorishniy et al., 2024) are particularly relevant to our work. For each data sample, these models retrieve information about other examples from the dataset, typically employing some form of attention mechanism (Bahdanau et al., 2014; Vaswani et al., 2017), and use it to make predictions. Thus, these models learn to find other relevant samples in the dataset, where relevance is determined by feature similarity. This can be viewed as an implicit learning of a similarity graph between data samples. A recent work by Liao & Li (2023) directly considers this as a problem of graph structure learning and applies a GNN on top of the learned graph. In contrast, in our work, we assume that some (ground-truth) relations between data samples are already given in advance, which is common in many real-world applications, and focus on the models that can utilize these relations.

**Machine learning for graphs** Graphs are a natural way to represent data from various domains. Hence, machine learning on graph-structured data has experienced significant growth in recent years, with Graph Neural Networks (GNNs) showing particularly strong results in many graph machine learning tasks. Most of the proposed GNN architectures (Kipf & Welling, 2016; Hamilton et al., 2017; Veličković et al., 2017) can be unified under a general Message-Passing Neural Networks (MPNNs) framework (Gilmer et al., 2017). However, GNNs are typically evaluated on graphs with homogeneous node features, most often text-based ones, such as bags of words or text embeddings. For instance, the most frequently used datasets for node classification are the three citation networks `cora`, `citeseer`, and `pubmed` (Sen et al., 2008; Namata et al., 2012; Yang et al., 2016; McCallum et al., 2000; Giles et al., 1998). The first two datasets use bags of words as node features, while the third one uses TF-IDF-weighted bags of words. Other datasets for node classification often found in the literature include coauthorship networks `coauthor-cs` and `coauthor-physics` (Shchur et al., 2018) that use bags of words as node features, and co-purchasing networks `amazon-computers` and `amazon-photo` (Shchur et al., 2018) that also use bags of words. In the popular Open Graph Benchmark (OGB) (Hu et al., 2020), `ogbn-arxiv`, `ogbn-papers100M`, and `ogbn-mag` datasets also use bags of words as node features, while `ogbn-products` uses dimensionality-reduced representations of bags of words. In a recently proposed benchmark of heterophilous graphs (Platonov et al., 2023b), `roman-empire` uses word embeddings as node features, while `amazon-ratings` and `questions` use bags of word embeddings. Recently, there has been a push for providing raw text descriptions of nodes for such text-attributed graphs (TAGs) in order to use models from the field of natural language processing (NLP) such as Large Language Models (LLMs) in combination with GNNs Chen et al. (2023b), but the features that GNNs receive in such scenarios are still homogeneous text embeddings. Several benchmarks providing such raw text descriptions of graph nodes have been created Khatua et al. (2023); Yan et al. (2023); Chen et al. (2024), and even a benchmark of graphs with both texts and images as node attributes has recently been proposed Zhu et al. (2024). While there are a few graph datasets that have heterogeneous tabular node features, such as `fraud-yelp` from Mukherjee et al. (2013) and `fraud-amazon` from McAuley & Leskovec (2013), they are not very popular in graph machine learning research, and the works that do use them (e.g., Zhang et al., 2020; Dou et al., 2020) typically do not apply any specialized feature processing and do not compare with GBDT baselines, thus ignoring the specifics of tabular node features. Indeed, fully leveraging the specifics of heterogeneous tabular node features for these datasets is difficult since feature types are not explicitly provided with the data. From these examples, it becomes clear that the effectiveness of graph machine learning models on graphs with heterogeneous tabular node features remains under-explored, despite such datasets being wide-spread in industry and science. We aim to partially address this issue with our benchmark and experimental results on it.

One more downside of existing popular GNN benchmarks is that they often do not provide realistic and meaningful prediction tasks. For instance, the most popular task in current academic graph machine learning is predicting paper subject areas in citation networks. However, this task can be better solved by analyzing the text of the paper with an LLM, or, even better, by simply using the subject area information provided by the paper authors. In contrast, we aim to provide datasets with meaningful prediction tasks.

**Machine learning for graphs with tabular node features**  Recently, two methods have been proposed specifically for learning on graphs with heterogeneous tabular node features. The first one is BGNN (Ivanov & Prokhorenkova, 2021), an end-to-end trained combination of GBDT and GNN, where GBDT takes node features as input and predicts node representations that are further concatenated with the original node features and used as input for a GNN. Another recent method is EBBS (Chen et al., 2022), which alternates between decision tree boosting and graph propagation steps, essentially fitting GBDTs to a graph-aware loss, and is also trained end-to-end. However, as Ivanov & Prokhorenkova (2021) note, there is currently a lack of publicly available datasets of graphs with heterogeneous node features. For this reason, the evaluation of these two methods provided in the original papers is limited, and the datasets used for it have various issues that make results obtained using them unreliable (see Appendix F for the detailed discussion). Thus, the community needs a better benchmark for evaluating models on graphs with heterogeneous tabular node features, and we aim to provide one.

One more research area related to our work is machine learning for relational databases (RDBs). In this field, graphs with heterogeneous tabular node features also appear, but the structure of these graphs is significantly different from that considered in our work. Specifically, in ML on RDBs, graphs are obtained from multi-table data with multiple relationship types between entities from different tables. In contrast, our work is focused on single-table data with a single type of relationships between entities in the same table. We discuss this area of research and its differences from our work in Appendix C. While representing data in different ways and thus using different graph ML methods, both ML for RDBs and our work aim to bring graph ML methods to tabular data, which we believe to be potentially a very fruitful direction.

## 3    TABGRAPHS: A BENCHMARK OF GRAPHS WITH TABULAR NODE FEATURES

In this section, we introduce our new benchmark of graph datasets with heterogeneous tabular node features. Our datasets cover the setting of transductive node property prediction (either node classification or node regression), which is the most common setting in modern graph machine learning and can be used to model many applications where tabular data appears. Some of these datasets were adapted from open sources (in this case, we either modified node features or added relational information to the datasets), while others are entirely new. Below, we briefly describe our datasets. A more detailed discussion, including the information about the sources of data, the preprocessing steps, and the presented node features, can be found in Appendix A.1. Note that in all our graphs, the edges are based on external information (e.g., inter-sample interactions, activity similarity, physical connections) rather than feature similarity, thus providing additional information for learning that is otherwise unavailable to models.

**tolokers-tab**  This dataset is based on the data provided by the Toloka crowdsourcing platform (Likhobaba et al., 2023). The nodes represent tolokers (workers) who perform work for customers, and they are connected by an edge if they have worked on the same task. The task is to predict which tolokers have been banned in one of the projects.

**questions-tab**  This dataset is based on the data from a question-answering website. The nodes represent users who post questions or leave answers on the site, and two users are connected by a directed edge if one of them has answered the other's question. The task is to predict which users remained active on the website (i.e., were not deleted or blocked).

**city-reviews**  This is a fraud detection dataset collected from a review service of organizations in two major cities. The nodes are users who leave ratings and post comments about various places, and they are connected with an edge if they have left reviews for the same organizations. The task is to predict whether a user leaves fraudulent reviews.

Table 1: Statistics of the proposed TabGraphs datasets.

| | node classification | | | | | | node regression | | | | |
|---|---|---|---|---|---|---|---|---|---|---|---|
| | tolokers-tab | questions-tab | city-reviews | browser-games | hm-categories | web-fraud | city-roads-M | city-roads-L | avazu-devices | hm-prices | web-traffic |
| # nodes | 11.8K | 48.9K | 148.8K | 15.2K | 46.5K | 2.9M | 57.1K | 142.3K | 76.3K | 46.5K | 2.9M |
| # edges | 519.0K | 153.5K | 1.2M | 5.1M | 10.7M | 12.4M | 107.1K | 231.6K | 11.0M | 10.7M | 12.4M |
| avg degree | 88.28 | 6.28 | 15.66 | 676.93 | 460.92 | 8.56 | 3.75 | 3.26 | 288.04 | 460.92 | 8.56 |
| % leaves | 3.6 | 53.1 | 25.9 | 6.1 | 8.6 | 48.4 | 0.1 | 0.1 | 5.6 | 8.6 | 48.4 |
| avg distance | 2.79 | 4.29 | 4.91 | 2.23 | 2.45 | 3.08 | 126.75 | 194.05 | 3.55 | 2.45 | 3.08 |
| diameter | 11 | 16 | 19 | 7 | 13 | 36 | 383 | 553 | 14 | 13 | 36 |
| global clustering | 0.23 | 0.02 | 0.26 | 0.47 | 0.27 | 0.00 | 0.00 | 0.00 | 0.24 | 0.27 | 0.00 |
| avg local clustering | 0.53 | 0.03 | 0.41 | 0.81 | 0.70 | 0.33 | 0.00 | 0.00 | 0.85 | 0.70 | 0.33 |
| degree assortativity | −0.08 | −0.15 | 0.01 | −0.40 | −0.35 | −0.14 | 0.70 | 0.74 | −0.30 | −0.35 | −0.14 |
| # classes | 2 | 2 | 2 | 16 | 21 | 2 | N/A | N/A | N/A | N/A | N/A |
| target assortativity | 0.09 | 0.02 | 0.59 | 0.05 | 0.08 | 0.01 | 0.74 | 0.72 | 0.18 | 0.12 | −0.21 |
| label informativeness | 0.01 | 0.00 | 0.31 | 0.03 | 0.02 | 0.00 | N/A | N/A | N/A | N/A | N/A |
| % labeled nodes | 100.0 | 100.0 | 93.3 | 100.0 | 100.0 | 100.0 | 63.3 | 86.8 | 100.0 | 100.0 | 99.7 |
| # num features | 6 | 19 | 11 | 33 | 1 | 109 | 6 | 6 | 4 | 0 | 109 |
| # cat features | 1 | 1 | 5 | 3 | 6 | 20 | 5 | 5 | 13 | 11 | 20 |
| # bin features | 2 | 11 | 38 | 0 | 0 | 139 | 15 | 15 | 0 | 0 | 139 |

**browser-games** This dataset is collected from an online game platform. The nodes represent browser games that are developed and uploaded by various independent publishers, and they are connected with an edge if they are frequently played by the same users during a specific period of time. The task is to determine the categories of games.

**hm-categories** and **hm-prices** These two datasets are obtained from a co-purchasing network of products from the H&M company (García Ling et al., 2022). The nodes are products and they are connected with an edge if they are frequently bought by the same customers. We prepared two datasets with the same graph but different tasks: for hm-categories, the task is to determine the categories of products, while for hm-prices, the task is to estimate their average prices.

**city-roads-M** and **city-roads-L** These datasets represent road networks of two major cities, with the second one being several times larger than the first. The nodes correspond to segments of roads, and a directed edge connects two segments if they are incident to each other and moving from one segment to another is permitted by traffic rules. The task is to predict the average traveling speed on the road segment at a specific timestamp.

**avazu-devices** This dataset is based on the data about user interactions with ads provided by the Avazu company (Wang & Cukierski, 2014). The nodes are devices used for accessing the internet and they are connected with an edge if they frequently visit the same websites. The task is to predict the click-through rate for devices based on data about viewed advertisements.

**web-fraud** and **web-traffic** These two datasets represent a part of the Internet. Here, nodes are websites, and a directed edge connects two nodes if at least one user followed a link from one website to another in a selected period of time. These weights on edges represent the number of users that moved between the websites. We prepared two datasets with the same graph but different tasks: for web-fraud, the task is to predict which websites are fraudulent, while for web-traffic, the task is to predict the logarithm of how many users visited a website on a specific day.

All graphs in our benchmark are (weakly-)connected graphs without self-loops. All graphs except for questions-tab, city-roads-M, city-roads-L, web-fraud, web-traffic are undirected, and all graphs except for web-fraud and web-traffic are unweighted. In our experiments, we convert all directed edges to undirected ones and do not use edge weights in order to run all experiments in a unified setting.

Some statistics of the proposed datasets are provided in Table 1 (note that we transformed all graphs to be undirected and unweighted for computing these statistics). We provide the definitions of these statistics and further discussion of the properties and diversity of our datasets in Appendix A.2. Overall, our datasets are diverse in domain, scale, structural properties, graph-label relations, and

node attributes. Providing meaningful prediction tasks, they may serve as a valuable tool for the research and development of machine learning models that can process graph-structured data with heterogeneous tabular node features.

# 4 SIMPLE MODEL MODIFICATIONS

Standard models for tabular machine learning cannot leverage information provided by the graph structure available in the data, while standard models for graph machine learning can struggle to efficiently use heterogeneous node features. Thus, both approaches can be suboptimal for learning on graphs with heterogeneous tabular node features. However, one can apply simple modifications to these models that can make tabular ML models graph-aware and enable graph ML models to better handle heterogeneous node features. Note that, while the modifications discussed in this section are quite simple, they were overlooked in previous works on learning on graphs with tabular node features, which focused on designing much more complicated models without comparing them to simple baselines.

## 4.1 MAKING TABULAR MODELS GRAPH-AWARE

A simple approach to make models that cannot explicitly process graphs perform better on graph-structured data is to augment node features with information obtained from the node's neighborhood in the graph. One way to do this is to aggregate the features of the node's (possibly multi-hop) neighbors and add this aggregated information to the node's original features. This method was popularized in modern graph machine learning literature by Wu et al. (2019). Their approach (SGC) aggregates features from the node's neighborhood by several passes of a standard graph convolution (Kipf & Welling, 2016) and uses these features instead of the original ones. We modify this approach to make it more suitable for heterogeneous features. In particular, we compute different kinds of feature statistics (such as mean, maximum, and minimum values) over the 1-hop neighborhood of each node and, together with the node degree, append them to the original features. Thus, the new features of each node contain the information about its neighborhood in the graph. We refer to this procedure as *Neighborhood Feature Aggregation* (NFA), since it mainly augments node features with information about its neighbors' features, and describe it in more detail in Appendix B.1. Note that this approach is very scalable, as it only requires a single pass over the graph edges to compute additional features for all nodes.

Other approaches to augmenting node features using graph structure are possible, for example, extending node features with information about local substructures in the graph or with node embeddings learned in an unsupervised way, such as DeepWalk (Perozzi et al., 2014) (see Appendix B.1 for more details). We experimented with some of these approaches and found that using DeepWalk node embeddings as additional node features provides significant performance gains on `city-roads-M` and `city-roads-L` datasets. Therefore, we use these node embeddings as additional features in all our experiments on these two datasets. One more important advantage of such feature augmentation approaches is that they allow tabular models to take advantage of unlabeled nodes in the graph, which are otherwise not used by these models, in contrast to GNNs.

## 4.2 MAKING GNNS WORK BETTER WITH HETEROGENEOUS FEATURES

The main problem of heterogeneous features for standard neural networks is the presence of numerical features. Even when preprocessed with transformations such as standard scaling, min-max scaling, or quantile transformation to standard normal or uniform distribution, these features still often cannot be used by neural networks as effectively as by decision trees (see Appendix B.2 for a discussion of this). However, Gorishniy et al. (2022) have recently proposed specialized numerical feature embeddings that can often improve the processing of numerical features by neural networks. This technique adds a learnable module that transforms numerical features into embeddings that are more suitable as input for neural networks. In particular, `Periodic-Linear-ReLU` (PLR) numerical feature embeddings tend to significantly improve the performance of neural networks working with numerical features, see Appendix B.2. However, this technique was never used with GNNs before. In our experiments, we apply PLR embeddings for training GNNs on graphs with heterogeneous node features.

## 5 EXPERIMENTS

In this section, we describe our experiments on the proposed TabGraphs benchmark. The details of our experimental setup and hyperparameter tuning for different models are provided in Appendix D.

### 5.1 MODELS

**Simple baseline**  As a simple baseline, we use a ResNet-like model: an MLP with skip-connections (He et al., 2016) and layer normalization (Ba et al., 2016). This model does not have any information about the graph structure and operates on nodes as independent samples (we call such models *graph-agnostic*). This model also does not have any specific design for working with tabular features.

**Tabular models**  We consider the three most popular implementations of GBDT: XGBoost (Chen & Guestrin, 2016), LightGBM (Ke et al., 2017), and CatBoost (Prokhorenkova et al., 2018). Further, we consider two recently proposed deep learning models for tabular data. One is MLP-PLR (Gorishniy et al., 2022), a simple MLP augmented with PLR numerical feature embeddings. It has been shown by Gorishniy et al. (2022) that this model outperforms many other tabular deep learning methods. Another is TabR-PLR (Gorishniy et al., 2024), which is a retrieval-augmented model. TabR-PLR also uses PLR embeddings for numerical features processing, although it uses a simplified version of them called *lite* (see Appendix B.2 for a detailed discussion), as is done in the original implementation of the model (Gorishniy et al., 2024). Note that all models discussed above are graph-agnostic.

To investigate how effectively graph structure can be used in combination with tabular models, we also experiment with the proposed NFA strategy for augmenting node features with information about the features of 1-hop neighbors in the graph, as described in Section 4.1. In particular, we provide such an augmentation for LightGBM, an efficient implementation of GBDT, and MLP-PLR, a simple yet strong neural baseline. We denote these models with -NFA suffix. Comparing the performance of standard models and their versions with graph-augmented node features is one way to see if graph information is helpful for the task.

**Graph deep learning models**  We also consider several representative GNN architectures. First, we use GCN (Kipf & Welling, 2016) and GraphSAGE (Hamilton et al., 2017) as simple classical GNN models. For GraphSAGE, we use the version with the mean aggregation function, and we do not use the neighbor sampling technique proposed in the original paper, instead training the model on a full graph, like all other GNNs in our experiments. Further, we use two GNNs with attention-based neighborhood aggregation functions: GAT (Veličković et al., 2017) and Graph Transformer (GT) (Shi et al., 2020). Note that GT is a *local* graph transformer, i.e., each node only attends to its neighbors in the graph (in contrast to *global* graph transformers, in which each node attends to all other nodes in the graph, and which are thus not instances of the standard MPNN framework). We equip all 4 considered GNNs with skip-connections and layer normalization, which we found important for stability and strong performance. We also add a two-layer MLP with the GELU activation function (Hendrycks & Gimpel, 2016) after every neighborhood aggregation block in GNNs. Our graph models are implemented in the same codebase as ResNet — we simply swap each residual block of ResNet with a residual neighborhood aggregation block of the selected GNN architecture. Therefore, comparing the performance of ResNet and GNNs is one more way to see if graph information is helpful for the task. Further, for all the considered GNNs, we experiment with their versions augmented with PLR embeddings, as described in Section 4.2 — we denote these models with -PLR suffix (we do not use these model modifications for the hm-prices dataset, since it does not contain numerical features).

**Specialized models**  We also use two recently proposed methods designed specifically for learning on graphs with heterogeneous tabular node features: BGNN (Ivanov & Prokhorenkova, 2021) and EBBS (Chen et al., 2022).

### 5.2 RESULTS

In this subsection, we compare and analyze the performance of the considered models on the proposed datasets. The results for classification and regression datasets are provided in Tables 2a and 2b, respectively.

Table 2: Experimental results. The best results are marked with orange, and the results for which the mean differs from the best one by no more than the sum of the two results' standard deviations are marked with cyan. For some experiments, the results are not reported due to one of the following reasons: `MLE` — memory limit exceeded, `TLE` — time limit exceeded, `FCP` — an experiment failed to outperform a constant prediction baseline, `N/A` — a separate experiment with PLR embeddings for numerical features is not needed since dataset does not have numerical features.

(a) Results for classification datasets. Accuracy is reported for multiclass datasets (`browser-games` and `hm-categories`), Average Precision (AP) is reported for binary classification datasets (all other datasets).

| | tolokers-tab | questions-tab | city-reviews | browser-games | hm-categories | web-fraud |
|---|---|---|---|---|---|---|
| ResNet | $45.17 \pm 0.61$ | $84.01 \pm 0.26$ | $64.33 \pm 0.32$ | $78.82 \pm 0.32$ | $70.45 \pm 0.24$ | $14.21 \pm 0.24$ |
| XGBoost | $48.79 \pm 0.25$ | $85.03 \pm 5.78$ | $65.55 \pm 0.18$ | $79.73 \pm 0.17$ | $71.08 \pm 0.70$ | $16.95 \pm 0.18$ |
| LightGBM | $48.49 \pm 0.27$ | $87.24 \pm 0.14$ | $66.17 \pm 0.17$ | $79.28 \pm 0.15$ | $71.09 \pm 0.10$ | $17.08 \pm 0.11$ |
| CatBoost | $48.61 \pm 0.25$ | $87.59 \pm 0.04$ | $66.05 \pm 0.16$ | $80.46 \pm 0.22$ | $71.14 \pm 0.12$ | TLE |
| MLP-PLR | $47.72 \pm 0.45$ | $87.34 \pm 0.42$ | $66.36 \pm 0.11$ | $80.69 \pm 0.24$ | $71.02 \pm 0.08$ | $16.24 \pm 0.12$ |
| TabR-PLR | $48.50 \pm 0.69$ | $85.56 \pm 0.52$ | $66.50 \pm 0.26$ | $80.29 \pm 0.26$ | $71.38 \pm 0.22$ | MLE |
| LightGBM-NFA | $57.99 \pm 0.43$ | $87.79 \pm 0.19$ | $71.66 \pm 0.11$ | $83.09 \pm 0.26$ | $81.72 \pm 0.12$ | $23.72 \pm 0.16$ |
| MLP-PLR-NFA | $57.70 \pm 0.20$ | $87.43 \pm 0.07$ | $71.93 \pm 0.12$ | $83.36 \pm 0.26$ | $81.35 \pm 0.21$ | $22.33 \pm 0.29$ |
| GCN | $61.09 \pm 0.38$ | $84.92 \pm 0.95$ | $71.08 \pm 0.32$ | $79.17 \pm 0.41$ | $86.42 \pm 0.31$ | $14.65 \pm 0.24$ |
| GraphSAGE | $57.08 \pm 0.24$ | $85.70 \pm 0.30$ | $71.15 \pm 0.27$ | $82.56 \pm 0.11$ | $86.35 \pm 0.18$ | $20.28 \pm 0.48$ |
| GAT | $58.77 \pm 1.00$ | $84.44 \pm 0.68$ | $71.38 \pm 0.53$ | $82.60 \pm 0.26$ | $87.84 \pm 0.23$ | $19.95 \pm 0.51$ |
| GT | $58.92 \pm 0.57$ | $83.59 \pm 1.17$ | $71.72 \pm 0.23$ | $83.29 \pm 0.33$ | $89.00 \pm 0.23$ | $20.19 \pm 0.44$ |
| GCN-PLR | $60.81 \pm 0.56$ | $88.80 \pm 0.25$ | $70.40 \pm 0.58$ | $80.50 \pm 0.58$ | $83.85 \pm 0.28$ | MLE |
| GraphSAGE-PLR | $60.28 \pm 0.97$ | $88.55 \pm 0.48$ | $72.26 \pm 0.40$ | $83.19 \pm 0.34$ | $86.77 \pm 0.12$ | MLE |
| GAT-PLR | $60.99 \pm 0.82$ | $88.69 \pm 0.63$ | $71.66 \pm 0.66$ | $83.59 \pm 0.33$ | $87.94 \pm 0.20$ | MLE |
| GT-PLR | $61.95 \pm 0.73$ | $82.41 \pm 1.60$ | $71.80 \pm 0.21$ | $83.26 \pm 0.36$ | $89.01 \pm 0.15$ | MLE |
| BGNN | $47.45 \pm 1.29$ | $47.20 \pm 5.24$ | $51.59 \pm 3.42$ | $75.92 \pm 0.34$ | $84.60 \pm 0.50$ | $3.21 \pm 0.15$ |
| EBBS | $43.86 \pm 1.63$ | $79.03 \pm 3.57$ | $57.40 \pm 1.99$ | $64.56 \pm 0.16$ | $41.77 \pm 1.89$ | $6.00 \pm 0.68$ |

(b) Results for regression datasets. $R^2$ is reported for all datasets.

| | city-roads-M | city-roads-L | avazu-devices | hm-prices | web-traffic |
|---|---|---|---|---|---|
| ResNet | $70.58 \pm 0.35$ | $67.49 \pm 0.09$ | $21.60 \pm 0.08$ | $67.31 \pm 0.21$ | $73.19 \pm 0.04$ |
| XGBoost | $71.54 \pm 0.08$ | $69.02 \pm 0.03$ | $24.43 \pm 0.03$ | $67.63 \pm 0.08$ | $75.41 \pm 0.01$ |
| LightGBM | $71.24 \pm 0.10$ | $68.86 \pm 0.08$ | $24.18 \pm 0.05$ | $68.52 \pm 0.15$ | $75.27 \pm 0.01$ |
| CatBoost | $71.88 \pm 0.09$ | $68.46 \pm 0.03$ | $25.56 \pm 0.12$ | $69.07 \pm 0.26$ | TLE |
| MLP-PLR | $71.52 \pm 0.17$ | $68.91 \pm 0.16$ | $22.52 \pm 0.29$ | $68.26 \pm 0.04$ | $74.53 \pm 0.04$ |
| TabR-PLR | $73.24 \pm 0.26$ | $72.92 \pm 0.07$ | MLE | $68.46 \pm 0.29$ | MLE |
| LightGBM-NFA | $72.59 \pm 0.10$ | $70.98 \pm 0.04$ | $31.97 \pm 0.03$ | $78.64 \pm 0.05$ | $86.66 \pm 0.01$ |
| MLP-PLR-NFA | $72.06 \pm 0.16$ | $68.81 \pm 0.13$ | $31.49 \pm 0.16$ | $75.18 \pm 0.50$ | $86.17 \pm 0.03$ |
| GCN | $72.87 \pm 0.21$ | $70.92 \pm 0.23$ | $27.31 \pm 0.17$ | $77.05 \pm 0.25$ | $81.95 \pm 0.08$ |
| GraphSAGE | $73.35 \pm 0.58$ | $71.03 \pm 0.90$ | $27.99 \pm 0.32$ | $76.01 \pm 0.47$ | $84.04 \pm 0.19$ |
| GAT | $73.64 \pm 0.30$ | $71.74 \pm 0.23$ | $28.28 \pm 0.54$ | $78.02 \pm 0.32$ | $84.85 \pm 0.17$ |
| GT | $72.95 \pm 0.47$ | $69.98 \pm 0.57$ | $30.27 \pm 0.26$ | $78.44 \pm 0.58$ | $85.17 \pm 0.17$ |
| GCN-PLR | $73.08 \pm 0.33$ | $70.95 \pm 0.18$ | $24.68 \pm 0.12$ | N/A | MLE |
| GraphSAGE-PLR | $73.51 \pm 0.37$ | $71.97 \pm 0.31$ | $27.64 \pm 0.23$ | N/A | MLE |
| GAT-PLR | $73.25 \pm 0.33$ | $71.78 \pm 0.20$ | $28.29 \pm 0.36$ | N/A | MLE |
| GT-PLR | $73.09 \pm 0.35$ | $71.12 \pm 0.56$ | $29.88 \pm 0.20$ | N/A | MLE |
| BGNN | $57.80 \pm 0.16$ | $58.73 \pm 0.34$ | $22.67 \pm 0.19$ | $70.23 \pm 0.56$ | FCP |
| EBBS | $24.40 \pm 3.06$ | $32.54 \pm 5.17$ | $8.59 \pm 1.73$ | $30.49 \pm 2.83$ | $26.39 \pm 0.16$ |

First, we note that our simplest baseline ResNet achieves worse results than GBDTs and neural tabular learning models on all the proposed datasets. This shows that our datasets indeed have meaningful tabular features, and methods designed specifically for tabular data outperform a simple deep learning approach.

Second, all the considered GNNs outperform ResNet on the proposed graph datasets, with the only exception of GT on the `questions-tab` dataset. Since our ResNet and GNNs are implemented in the same codebase and are hence directly comparable, this shows that the graph structure in our datasets is indeed helpful for the given tasks.

Further, we can see that the best of vanilla GNNs outperforms the best of graph-agnostic models on all of the proposed datasets, except for `city-roads-L`. We also note that these differences in performance are often quite large. For example, on `hm-categories`, the best model without any knowledge about the graph structure reaches only 71.38 points of Accuracy, while the best vanilla GNN achieves 89.00 points.

The relative performance of different GNN models varies across graph datasets, and there is no best GNN architecture that works universally well for each of the datasets. For instance, GCN appears to be the best on `tolokers-tab`, while GraphSAGE is the best on `quesions-tab`,

GAT is the best on `city-roads-M`, and GT is the best on `hm-categories`. Thus, it is hard to make a prior choice of architecture, and comparative experiments are required for each dataset. While the differences in performance between different GNN architectures are not always large, it is important to note that, for industrial applications, such as fraud detection or CTR prediction, even small improvements can be very important.

Further, we can notice that BGNN and EBBS, the recently proposed models designed specifically for graphs with heterogeneous tabular node features, failed to produce results competitive with vanilla GNNs or even graph-agnostic tabular models on almost all of our datasets. We hypothesize that these models were over-engineered to achieve strong performance on a particular set of mostly small and flawed datasets used in previous works (see Appendix F for a discussion of these flaws), and their results do not transfer to more realistic datasets.

Now, let us discuss the performance of the simple model modifications proposed in Section 4. First, we note that PLR embeddings for numerical features often improve the performance of GNNs, and sometimes these improvements are quite large. For example, on `questions-tab`, GCN achieves 84.92 points of AP, while its augmented version GCN-`PLR` reaches 88.80 points. Further, such a simple feature augmentation as aggregating feature information over 1-hop graph neighbors leads to significant performance gains for the considered tabular models, which are originally graph-agnostic. For instance, on `tolokets-tab`, LightGBM achieves 48.49 points of AP, while its version with graph-augmented features LightGBM-`NFA` reaches 57.99 points. Similarly, on `hm-categories`, MLP-`PLR` achieves 71.02 points of Accuracy, while MLP-`PLR-NFA` reaches 81.35 points. Overall, graph-augmented tabular models provide the best results on 4 of the considered datasets. This shows that, when simply provided with feature augmentation based on the local graph structure, standard tabular models become strong baselines and can sometimes compete with and even outperform GNNs, which has been overlooked in previous studies.

Based on our results, we obtained the following insights that we hope will be useful for researchers and practitioners working with tabular and graph data:

- If it is possible to define meaningful relations between data samples, it is worth trying to convert the given data to a graph and experiment with ML methods that are capable of processing graph information, as it can lead to significant performance gains.
- Standard GNNs can provide strong performance on graphs with tabular node features, but the best graph neural architecture depends on the specific dataset, and it is important to experiment with different design choices.
- The recently proposed models designed specifically for graphs with heterogeneous node features are consistently outperformed by standard tabular and graph ML models.
- The recently proposed PLR embeddings for numerical features can be easily integrated into GNNs and further improve their performance in many cases.
- Graph-based feature augmentation allows graph-agnostic tabular models to use relational information in the graph and significantly improves their performance, making them a strong and simple baseline. Such models are easy to experiment with in existing tabular ML pipelines, so we recommend using them for initial experiments to check if graph structure is helpful in a particular application.

## 6 CONCLUSION

To conclude, we introduce TabGraphs, a benchmark for learning on graphs with heterogeneous tabular node features, which covers various industrial applications and includes graphs with diverse properties and meaningful prediction tasks. Using the proposed benchmark, we evaluate a large set of models, including standard baselines and advanced neural methods for both tabular and graph-structured data. Our experiments show that several previously overlooked model modifications, such as node feature augmentation based on graph neighborhood for graph-agnostic tabular models or numerical feature embeddings for GNNs, allow one to achieve the best performance on such data. Based on our empirical study, we provide insights and tips for researchers and practitioners in both tabular and graph machine learning domains. We hope that the proposed datasets will contribute to further developments in these fields by encouraging the use of graph ML methods for tabular data and by providing an alternative testbed for evaluating models for learning on graph-structured data.

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

# A  THE TABGRAPHS BENCHMARK DETAILS

## A.1  DATASETS DESCRIPTIONS

In this section, we provide more detailed descriptions of the TabGraphs datasets. The instruction on how to download the datasets can be found in our anonymous repository. In most of our datasets, the features come with their names, which are stored in our data files (the exceptions are the `city-reviews`, `browser-games`, `web-fraud`, and `web-traffic` datasets, for which the features are anonymized). Note that none of the proposed datasets contain any personal information.

**tolokers-tab**   This is a new version of the `tolokers` dataset from Platonov et al. (2023b). It is based on the data from the Toloka crowdsourcing platform (Likhobaba et al., 2023). The nodes represent tolokers (workers) and they are connected by an edge if they have worked on the same task within one of several projects. The task is to predict which tolokers have been banned in one of the projects. For this dataset, we kept the task and graph from the previous version, but replaced the processed node features with unprocessed ones. The new node features include various performance statistics of workers, such as the number of approved assignments and the number of skipped assignments (numerical features), as well as worker's profile information, such as their education level (categorical feature).

**questions-tab**   This is a new version of the dataset `questions` from Platonov et al. (2023b). It is based on the data from a question-answering website. Here, nodes represent users, and a directed edge $(u, v)$ connects users $u$ and $v$ if user $u$ answered a question posted by user $v$. The task is to predict which users remained active on the website (i.e., were not deleted or blocked). For this dataset, we kept the task and graph from the previous version (except we provide directed edges, in contrast to the previous version of the dataset in which the graph was converted to an undirected one and no information about the original edge directions was provided), but replaced the node features to make them heterogeneous. The original version of this dataset used bag of word embeddings representations of user descriptions as node features, while our features are based on the activity of users on the website, such as articles count, achievements count, subscribers count, categories subscriptions count, rating (numerical features), as well as their profile information, such as what city the user is from (categorical feature), whether the user has a profile description, whether the user has filled the education field, whether the user has left a contact URL (binary features). Note that these new features are much more predictive of the target, as demonstrated by much better performance achieved by models on the new version of the dataset compared to the previous one.

**city-reviews**   This dataset is obtained from the logs of a review service. It represents the interactions between users of the service and various organizations located in two major cities. The organizations are visited and rated by users, so the dataset is originally bipartite and contains entities of these two types. Thus, we transform it by projecting to the part of users. Let $\mathbf{P} \in \{0, 1\}^{m \times n}$ be a binary adjacency matrix of users and organizations, where $m$ is the number of organizations, $n$ is the number of users, and $p_{ij}$ denotes whether a user $j$ has left a review for an organization $i$. Then, $\mathbf{B} = \mathbf{P}^\top \mathbf{P} \in \mathbb{R}^{n \times n}$ corresponds to the weighted adjacency matrix of users, where $b_{ij}$ is the dot product of columns $i$ and $j$ in $\mathbf{P}$. Here, the more common rated organizations there are for two users, the greater the weight of the connection between them. Further, we obtain a binary adjacency matrix $\mathbf{A} \in \{0, 1\}^{n \times n}$ of users with $a_{ij} = [b_{ij} \geqslant \gamma]$ by applying a threshold $\gamma$ to the weights $b_{ij}$. The resulting graph is undirected, and the task is to predict whether a user is a fraudster. The features include the information about the user profile, such as the length of the nickname in characters (numerical feature) and whether the profile information is hidden (binary feature), as well as their behavior on the websites and other services, such as the share of negative ratings among user reviews, the number of search queries, the number of different categories in search queries (numerical features), the type of browser (categorical feature).

**browser-games**   This dataset is obtained from the logs of an online game platform and represents the network of browser games that are created and hosted by various game developers. These games are played by users, so the dataset is originally bipartite and contains entities of these two types. Thus, we transform it by projecting to the part of games. Let $\mathbf{P} \in \{0, 1\}^{m \times n}$ be a binary adjacency matrix of users and games, where $m$ is the number of users, $n$ is the number of games, and $p_{ij}$

denotes whether a user $i$ has played a game $j$. Then, $\mathbf{B} = \mathbf{P}^\top \mathbf{P} \in \mathbb{R}^{n \times n}$ corresponds to the weighted adjacency matrix of games, where $b_{ij}$ is the dot product of columns $i$ and $j$ in $\mathbf{P}$. The more common users there are for the given pair of games, the greater the weight of the connection between these games. Further, we obtain a binary adjacency matrix $\mathbf{A} \in \{0, 1\}^{n \times n}$ of games with $a_{ij} = [b_{ij} \geqslant \gamma]$ by applying a threshold $\gamma$ to the weights $b_{ij}$. The resulting graph is undirected, and the task is to predict the categories of games. The features describe various attributes of games, such as the average play time on different platforms (numerical feature), the most popular user language (categorical feature), game retention (numerical feature), the number of user clicks (numerical feature), various information about the game publisher and many other game statistics (numerical features).

**hm-categories** and **hm-prices**   These datasets are based on an open-source dataset that has been introduced at the Kaggle competition hosted by H&M Group (García Ling et al., 2022). The dataset is originally bipartite and contains entities of two types — customers and products that they purchase at the H&M shop. Thus, we transform it by projecting to the part of products. The connections in the original dataset can be described by a weighted adjacency matrix $\mathbf{P} \in \mathbb{R}^{m \times n}$, where $m$ is the number of users, $n$ is the number of products, and $p_{ij}$ denotes how many times a user $i$ has bought a product $j$. Then, $\mathbf{B} = \mathbf{P}^\top \mathbf{P} \in \mathbb{R}^{n \times n}$ corresponds to the weighted adjacency matrix of products, where $b_{ij}$ is the dot product of columns $i$ and $j$ in $\mathbf{P}$. The more often either of two products is bought by a common customer, and the more shared customers there are in general, the greater the weight $b_{ij}$ of the connection between these products. After that, we obtain a binary and more sparse adjacency matrix $\mathbf{A} \in \{0, 1\}^{n \times n}$ of products with $a_{ij} = [b_{ij} \geqslant \gamma]$ by applying a threshold $\gamma$ to the weights $b_{ij}$. The resulting graph is undirected. For this dataset, we consider two different versions: hm-categories with the product group as the target for the classification task and hm-prices with the average price of a product as the target for the regression task. In both cases, we adjust the set of features so that the problem does not become trivial, but the underlying graph is the same for these two versions. For the regression task, we consider such features as product types, graphical appearance (categorical feature), perceived color (categorical feature), etc. For the classification task, the set of features includes average price (numerical feature) and a reduced subset of categorical attributes from the regression task, which makes the problem more challenging.

**city-roads-M** and **city-roads-L**   These datasets are obtained from the logs of a navigation service and represent the road networks of two major cities. Here, city road segments are considered as graph nodes, and there is a directed edge from node $i$ to node $j$ if the corresponding road segments are incident to each other and moving from $i$ to $j$ is permitted by traffic rules. Thus, the obtained graph is directed, and we extract its largest weakly connected component. The task is to predict the travel speed on roads at a specific timestamp. The features include numerous binary indicators describing a road, such as whether there is a bike dismount sign, whether the road segment ends with a crosswalk or toll post, whether it is in poor condition, whether it is restricted for trucks or has a mass transit lane. Other features include the length of the road and the geographic coordinates of the start and the end of the road (numerical features), as well as the speed mode of the road (categorical feature). For these datasets, we found that providing DeepWalk node embeddings (Perozzi et al., 2014) as additional input node features to different models significantly improves their performance, so we use these embeddings in all our experiments with these datasets and provide them with the data.

**avazu-devices**   This is another dataset based on open-source data that has been introduced at the Kaggle competition organized by Avazu (Wang & Cukierski, 2014). It represents the interactions between devices and advertisements on the internet. This dataset is originally bipartite and contains entities of three types — devices, websites that are visited by these devices, and applications that are used to visit them. A version of this dataset has been used by Ivanov & Prokhorenkova (2021) in their study. However, it contained only a small subset of interactions from the original dataset, which resulted in a small-sized graph. Because of that, we decided to consider the whole dataset and transform it by projecting to the part of devices. Let $\mathbf{P} \in \mathbb{R}^{m \times n}$ be a weighted adjacency matrix of devices and entry points defined as the combinations of sites and applications, where $m$ is the number of entry points, $n$ is the number of devices, and $p_{ij}$ denotes how many times device $j$ has used entry point $i$ (i.e., visited a specific site under a specific application). Then, $\mathbf{B} = \mathbf{P}^\top \mathbf{P} \in \mathbb{R}^{n \times n}$ corresponds to the weighted adjacency matrix of devices, where $b_{ij}$ is the dot product of columns $i$ and $j$ in $\mathbf{P}$. The interpretation of this matrix is similar to what we discussed above for hm-products.

Finally, we obtain a binary adjacency matrix $\mathbf{A} \in \{0,1\}^{n \times n}$ of devices with $a_{ij} = [b_{ij} \geqslant \gamma]$ by applying a threshold $\gamma$ to the weights $b_{ij}$. The resulting network is undirected. The task is to predict the click-through rate (CTR) observed on devices. The features include numerous categorical attributes, such as device model, banner position, and some number of additional features that have been already anonymized before being released to public access.

**web-fraud** and **web-traffic**   These two datasets represent a segment of the Internet. Here, websites are treated as nodes, and there is a directed edge from node $i$ to node $j$ with weight $w_{ij}$, if there were $w_{ij}$ users who followed a link between websites $i$ and $j$ in a selected period of time. We prepared two datasets with the same graph but different tasks: for web-fraud, the task is to predict which websites are fraudulent, while for web-traffic, the task is to predict the logarithm of how many users visited a website on a specific day. The features in the dataset were obtained from the website content, such as the numbers of incoming and outgoing links, the numbers of words and sentences in the text content, the number of videos on the website (numerical features), the website's zone and what topic is the website dedicated to based on a classifier's prediction (categorical features), whether the website is on a free hosting and whether it has numbers in its address (binary features).

## A.2   DATASETS PROPERTIES

A key characteristic of our benchmark is its diversity. As described above, our graphs come from different domains and have different prediction tasks. Their edges are also constructed in different ways (based on user interactions, activity similarity, physical connections, etc.). However, the proposed datasets also differ in many other ways. Some properties of our graphs are presented in Table 1 (see below for the details on how the provided statistics are defined). First, note that the sizes of our datasets range from 11K to 3M nodes. The smaller graphs can be suitable for compute-intensive models, while the larger graphs can provide a moderate scaling challenge. The average degree of our graphs also varies significantly — most graphs have the average degree ranging from tens to hundreds, which is larger than the average degrees of most datasets used in present-day graph ML research; however, we also have some sparser graphs such as questions-tab, city-roads-M, city-roads-L. The average distance between two nodes in our graphs varies from 2.23 for browser-games to 194 for city-roads-L, and graph diameter (maximum distance) varies between 7 for browser-games to 553 for city-roads-L. Further, we report the values of clustering coefficients which show how typical are closed node triplets for the graph. In the literature, there are two definitions of clustering coefficients (Boccaletti et al., 2014): the global clustering coefficient and the average local clustering coefficient. We have graphs where the clustering coefficients are high or almost zero, and graphs where global and local clustering coefficients significantly disagree (which is possible for graphs with imbalanced degree distributions). The degree assortativity coefficient is defined as the Pearson correlation coefficient of degrees among pairs of linked nodes. Most of our graphs have negative degree assortativity, which means that nodes tend to connect to other nodes with dissimilar degrees, while for the city-roads-M and city-roads-L datasets the degree assortativity is positive and large.

Further, let us discuss the graph-label relationships in our datasets. To measure the similarity of labels of connected nodes for regression datasets, we use target assortativity — the Pearson correlation coefficient of target values between pairs of connected nodes. For instance, for the city-roads-M and city-roads-L datasets, the target assortativity is positive and quite large, which shows that nodes tend to connect to other nodes with similar target values, while for the web-traffic dataset, the target assortativity is negative. For classification datasets, the similarity of neighbors' labels is usually called *homophily*: in homophilous datasets, nodes tend to connect to nodes of the same class. We use adjusted homophily to characterize homophily level, as it has been shown to have more desirable properties than other homophily measures used in the literature (Platonov et al., 2023a). In Table 1, we refer to adjusted homophily as *target assortativity*, as it is a special case of the assortativity coefficient (Newman, 2003). We can see that for the city-reviews dataset, adjusted homophily is positive and quite large, which shows that this dataset is homophilous, while for the rest of our classification datasets, adjusted homophily is close to zero. One more characteristic to describe graph-label relationships is label informativeness (Platonov et al., 2023a). It shows how much information about the label of a given node can be derived from the label of a neighbor node. In our datasets, label informativeness correlates with adjusted homophily, which is typical for real-world labeled graphs.

Note that some of our graphs contain unlabeled nodes. This is a typical situation for industry and science, yet it is underrepresented in graph machine learning benchmarks. Unlabeled nodes give an additional advantage to graph-aware models, as they can utilize the information about the features of these nodes and their position in the graph even without knowing their labels.

Finally, our datasets have sets of heterogeneous tabular node features with different number and balance of numerical, categorical, and binary attributes. The numerical features have widely different scale and distribution. For example, for the `questions-tab` dataset, most of the features are counters (questions count, answers count, subscribers count, achievements count, articles count) with different scales, while the rating feature has a very different distribution with negative values and lots of outliers.

Overall, our datasets are diverse in domain, scale, structural properties, graph-label relations, and node attributes. Providing meaningful prediction tasks, they may serve as a valuable tool for the research and development of machine learning models that can process graph-structured data with heterogeneous features.

**Computing dataset statistics**  Let us further describe the statistics that we use in Table 1. Note that before computing all the considered graph characteristics, we transformed the graphs to be undirected and unweighted, since some of the characteristics are only defined for such graphs.

*Average degree* is the average number of neighbors a node has. *% leaves* is the percentage of nodes of degree 1 in the graph. Since all our graphs are connected (when treated as undirected graphs), for any two nodes there is a path between them. *Average distance* is the average length of the shortest paths among all pairs of nodes, while *diameter* is the maximum length of the shortest paths among all pairs of nodes. For our largest graph — the one used for `web-fraud` and `web-traffic` datasets — we approximate average distance with an average over distances for 100K randomly sampled node pairs. *Global clustering* coefficient is computed as the tripled number of triangles divided by the number of pairs of adjacent edges (i.e., it is the fraction of closed triplets of nodes among all connected triplets). *Average local clustering* coefficient first computes the local clustering of each node, which is the fraction of connected pairs of its neighbors, and then averages the obtained values among all nodes. *Degree assortativity* is the Pearson correlation coefficient between the degrees of connected nodes. Further, *target assortativity* for regression datasets is the Pearson correlation coefficient between target values of connected nodes. For classification tasks, we measure target assortativity via *adjusted homophily* (Platonov et al., 2023a) that can be computed as follows:

$$h_{\text{adj}} = \frac{h_{\text{edge}} - \mu}{1 - \mu}, \text{ with } \mu = \sum_{k=1}^{C} D_k^2/(2|E|)^2 \text{ and } D_k = \sum_{v \,:\, y_v = k} d_v,$$

where $h_{\text{edge}}$ is the fraction of edges connecting nodes with the same label, and $d_v$ denotes the degree of a node $v$. In Platonov et al. (2023a), it was shown that adjusted homophily satisfies a number of desirable properties, which makes it appropriate for comparing datasets with different number and balance of classes. Finally, *label informativeness* (LI) was introduced by Platonov et al. (2023a) and was shown to be more consistent with GNN performance than homophily. Label informativeness quantifies how much information a neighbor's label gives about the node's label. To formally define this measure, we let $(\xi, \eta) \in E$ be an edge sampled uniformly at random among all edges and define

$$\text{LI} := I(y_\xi, y_\eta)/H(y_\xi).$$

Here $y_\xi$ and $y_\eta$ are (random) labels of $\xi$ and $\eta$, $H(y_\xi)$ is the entropy of $y_\xi$ and $I(y_\xi, y_\eta)$ is the mutual information of $\xi$ and $\eta$.

# B  SIMPLE MODIFICATIONS FOR TABULAR MODELS AND GNNS

## B.1  FEATURE AUGMENTATION BASED ON GRAPH STRUCTURE

There are a number of possible approaches to augmenting node features with graph-based information in order to provide graph-agnostic models with some information about the graph.

**Neighborhood Feature Aggregation (NFA)**  First, we describe our Neighborhood Feature Aggregation (NFA) technique. This technique augments node features with the information about

features of the node's neighbors in the graph. As we show in our experiments, this technique often significantly improves the performance of graph-agnostic tabular models. We consider the set of 1-hop neighbors of each node and compute various statistics over the node features in this set. In particular, for numerical features, we compute their mean, maximum, and minimum values in the neighborhood. For categorical features, we first transform them into a set of binary features using one-hot encoding. Then, for each binary feature, be it an original binary feature or a binary obtained from a categorical one by one-hot encoding, we compute their mean values in the neighborhood, i.e., their ratios of 1s for binary indicators. Additionally, we compute the degree of the node and use it as one more additional feature. Then, we concatenate all the produced additional features with the original node features. We treat all these additional features as numerical features in our experiments, i.e., apply scaling transformations and possibly PLR embeddings to them.

Let us describe the NFA procedure slightly more formally. Let us consider some specific feature $x \in X$ from the set of features $X$, an arbitrary node $v \in V$ in the graph $G(V, E)$ and its 1-hop neighbors $\mathcal{N}_G(v)$. Then, we can collect the values of this feature for the node $v$ and its neighbors $\mathcal{N}_G(v)$ and apply some aggregation function $f$ (e.g., `mean`, `max`, `min`) to them in order to obtain a single value $h$:

$$h = f\Big(x_v, \big\{x_u : u \in \mathcal{N}_G(v)\big\}\Big).$$

This value $h$ is then used as an additional feature for the considered node $v$. This procedure is done for each node $v \in V$ and each feature $x \in X$. In particular, for numerical features, we apply three aggregation functions separately: `mean`, `max`, `min`, thus producing three new features. For binary features, we apply the `mean` aggregation function, hence producing one new feature. For categorical features, we first apply one-hot encoding to them, and then apply the `mean` aggregation function to each of the resulting binary features, thus producing as many additional features as there were possible values of the original categorical feature. In addition, we also append the node degree to the resulting NFA vector. We concatenate this NFA vector to the original features of the node $v$. We treat all these additional node features as numerical.

For example, consider the `questions-tab` dataset from our benchmark. Each node has a numerical feature `answers_count` — the number of answers the user represented by this node has given to questions asked on the question-answering platform. Based on this feature, our NFA procedure creates three additional numerical node features: `answers_count_mean`, `answers_count_max`, `answers_count_min`, which for each node contain respectively the mean, maximum, and minimum of the values of the `answers_count` feature for all 1-hop neighbors of the node in the graph. Each node also has a binary feature `has_description` — an indicator if the user has provided a profile description. Based on this feature, our NFA procedure creates one additional numerical node feature — `has_description_mean`. Each node also has a categorical feature `profile_quality` characterizing the quality of the user profile estimated by a model. This feature has four possible values encoded as integers 0, 1, 2, 3. Based on this feature, our NFA procedure creates four additional numerical features: `profile_quality_is_0_mean`, `profile_quality_is_1_mean`, `profile_quality_is_2_mean`, `profile_quality_is_3_mean`.

In Figure 1 we provide a simple illustration of our approach. Here, we consider a central node with three neighbors, which have one numerical feature (blue), one categorical feature (green), and one binary feature (orange). To construct NFA for the central node, we compute across all its neighboring nodes (including the central node, as it has a self-loop) the minimum, maximum, and average values for the numerical feature, average values for the one-hot-encoded categorical feature, and average value for the binary feature. After that, we append the node degree to the NFA vector. Note that, if categorical features are present, we first transform them to binary features with one-hot-encoding, so NFA is always applied to numerical or binary features.

**Other possible approaches to graph-based feature augmentation** It is possible to augment node features with other types of information obtained from the graph structure besides aggregating neighborhood node features. For example, counters of local graph substructures (e.g., network motifs or, more generally, graphlets) or various node centrality values can be used to extend node features. Further, different self-supervised node embeddings can be added to node features. We experimented with these approaches and found that using DeepWalk node embeddings (Perozzi et al., 2014) is very beneficial on `city-roads-M` and `city-roads-L` datasets, and thus use them in all our

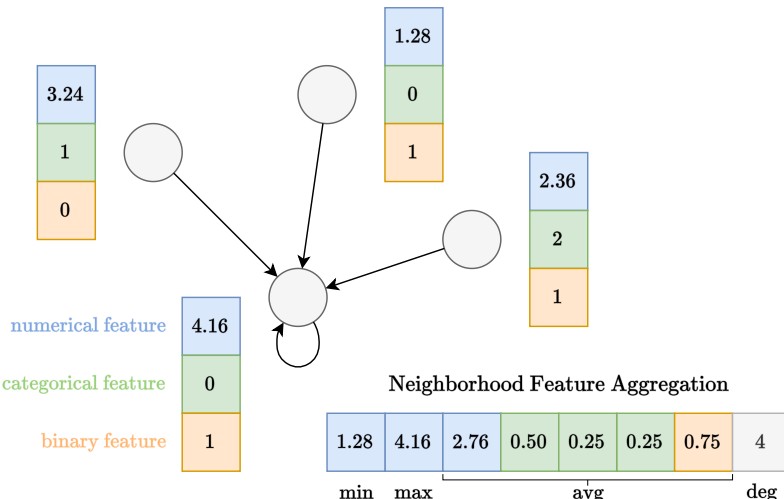

Figure 1: An illustrative example of applying Neighborhood Feature Aggregation (NFA).

experiments with these two datasets. Note that these datasets (which represent road networks) are originally embedded in two-dimensional space, and we hypothesize that DeepWalk embeddings combine information about graph connectivity with implicitly learned information about node relative positions in this space, which explains why these node embeddings are helpful. It is particularly interesting that DeepWalk node embeddings provide additional benefits to models even despite the fact that the coordinates of starts and ends of road segments (nodes) are provided in node features in `city-roads-M` and `city-roads-L` datasets, so models can leverage them to obtain similar information. While there are many other methods for obtaining unsupervised node embeddings and DeepWalk is a relatively old one among them, in our experiments we chose to focus on DeepWalk because, as recent research shows, it often performs on par with or even better than methods proposed after it (Gurukar et al., 2022).

### B.2 EMBEDDINGS FOR NUMERICAL FEATURES

In practice, neural networks are good at handling data with binary features and with categorical features transformed to binary features by one-hot encoding (when input features are immediately followed by a linear transformation, which is typically the case in neural networks, the model essentially learns an embedding for each binary feature). However, numerical features can be a problem for neural networks. Standard neural network building blocks represent (mostly) smooth functions, and thus neural networks cannot sharply vary their predictions based on changes in numerical features. However, in many cases, such smooth decisions are suboptimal. In contrast, decision trees, which are the basis of GBDT, select thresholds for values of numerical features and make hard discontinuous decisions based on them. This strategy often fits tabular data with rich numerical features better, and this is often considered to be one of the main reasons why neural networks still cannot consistently outperform GBDT models on tabular data, despite the large amount of research resources spent on improving neural networks for tabular data (Shwartz-Ziv & Armon, 2022; Grinsztajn et al., 2022; McElfresh et al., 2023).

Recently, Gorishniy et al. (2022) proposed several specialized techniques for embedding numerical features to improve the performance of neural networks on tabular data. These techniques introduce learnable modules that transform numerical features into embeddings — arguably a more natural form of inputs for neural networks. Of several methods proposed in the original paper, we focus on `Periodic-Linear-ReLU` (PLR) numerical feature embeddings, since according to the experiments from the original paper they provide the best average performance. This method is inspired by periodic activation functions that have recently found success in computer vision (Mildenhall et al., 2021; Tancik et al., 2020; Sitzmann et al., 2020; Li et al., 2021). The PLR embedder first passes a numerical feature through several sine and cosine functions with different (learnable) frequencies obtaining a periodic embedding, and then passes this embedding through a linear layer and a ReLU

activation function obtaining the final numerical feature embedding that becomes the input to the main model (see the original paper of Gorishniy et al. (2022) for more details).

Note that later a more memory-efficient modification of PLR embeddings called *lite* was introduced by (Gorishniy et al., 2024). This method differs from the original PLR embeddings in that, in order to save memory, it uses a shared linear layer for all input numerical features instead of a unique linear layer per feature. We found that the original PLR embeddings often perform better than PLR-lite embeddings, so, in our work, we use the original PLR embeddings for all experiments that include numerical feature embeddings, except for experiments with the TabR model for which PLR-lite embeddings were originally introduced. For this model, we follow the official implementation and use PLR-lite embeddings.

## C  ADDITIONAL RELATED WORK: GRAPH MACHINE LEARNING FOR RELATIONAL DATABASES

A field of research related to our work is machine learning for relational databases. This field also deals with a combination of tabular and graph data, although these types of data appear in it in a form different from the one studied in our work. In this field, the data is represented as a relational database: a collection of tables, each containing objects of a single type, with specified relationships between entries of different tables. Such data can be represented as a *heterogeneous* graph: a graph with multiple types of nodes and/or edges.[1] The nodes correspond to table entries (with nodes from different tables having different types) and edges correspond to relationships between entries of different tables (with different types of edges representing different kinds of relationships). Due to this possibility of representing relational databases as a heterogeneous graph, there have been several works applying graph machine learning methods to relational databases (Schlichtkrull et al., 2018; Cvitkovic, 2020; Krivosheev et al., 2020; Bai et al., 2021; Vogel et al., 2023; Zahradník et al., 2023; Hilprecht et al., 2023; Zhang et al., 2023). Regarding publicly available datasets, for a long time the main source of open RDB data for machine learning was the Prague Relational Learning Repository (Motl & Schulte, 2015). However, some of its datasets are synthetic, most of its datasets are quite small, and not all of its tasks are realistic. Further, on some of its tasks, even quite simple models achieve nearly perfect performance, hence these tasks cannot be used for meaningful model comparison (Wang et al., 2024). Later, several temporal RDB datasets for machine learning were introduced in the KDD Cup 2019 AutoML Competition (Zhou et al., 2020). However, these datasets, while being obtained from real-world industrial applications, do not provide any information about feature and target names or even data domains, which makes working with them particularly difficult. Very recently, two benchmarks of large-scale relational databases have been proposed: RelBench (Fey et al., 2023) and 4DBInfer (Wang et al., 2024).

Machine learning over relational databases is related to our work because it often also aims to bring graph ML methods to tabular data. Entities in relational database tables (and nodes in the corresponding heterogeneous graph) typically have heterogeneous tabular features, so using methods that can effectively work with graph structure and heterogeneous features is desirable. However, relational database ML differs from our work in the structure of graphs used to represent relational databases. In a relational database, there are several tables, and relationships usually can exist only between entities from different tables. Therefore, a relational database can be modeled with a heterogeneous graph where nodes of the same type usually cannot be connected (i.e., this heterogeneous graph is typically also multipartite). In contrast, in our work, we focus on datasets that consist of a single table (which is the standard setting for tabular machine learning) and have additional information about relationships between entities in this table (and all of these relationships are of the same type, although in general this need not be the case even for single-table data). Thus, such datasets can be modeled with a homogeneous graph (i.e., a graph in which all nodes have the same type and all edges have the same type). Taking into account that working with heterogeneous graphs requires specialized methods that can be quite different from those used for homogeneous graphs, we consider that different benchmarks and possibly different methods are needed for machine learning over relational databases

---

[1]Note that there is a difference between heterogeneous graphs and graphs with heterogeneous node features. The first term refers to heterogeneity of the structure of the graph, while the second term refers to heterogeneity of the types of node features. Graphs with heterogeneous node features can be either homogeneous (which is the focus of our work) or heterogeneous (which appear in machine learning over relational databases).

and machine learning over single-table tabular data with additional relational information. Both types of data are widespread in industry and science and have attracted attention from machine learning researchers and practitioners. The purpose of our work is to create a benchmark of single-table tabular datasets with additional relational information since there is currently a lack of open datasets of this type. That being said, both machine learning for relational databases and our work aims to bring graph machine learning methods to tabular data, which we believe to be potentially a very fruitful direction.

## D  EXPERIMENTAL SETUP AND HYPERPARAMETER SELECTION DETAILS

For our experiments, we split nodes of each dataset into train, validation, and test sets in proportion $50\%/25\%/25\%$. These splits are random and stratified, where the stratification is done by class for the classification datasets and by target variable quantile for the regression datasets. The only exception is `city-reviews`, where we choose a natural split based on the city. Recall that the organizations reviewed by users (nodes) in this dataset are located in two cities, so we can split the users into two groups based on in which city most of the organizations they were interacting with are located. Then, the users from the larger group are split randomly into train and validation sets in proportion $50\%/50\%$, while the users from the smaller group form the test set. The resulting proportion for train/validation/test splits for the `city-reviews` dataset becomes approximately $34\%/34\%/32\%$. This split introduces a natural distribution shift between train/validation and test sets. We report Average Precision (AP) for binary classification datasets, Accuracy for multiclass classification datasets, and $R^2$ for regression datasets. For each dataset, we train each model 5 times with different random seeds and report the mean and standard deviation of performance in these runs.

Some of the graphs in our benchmark are directed and/or weighted. In order to run all experiments in a unified setting, we converted directed graphs to undirected ones (by replacing each directed edge with an undirected one) and did not use edge weights in weighted graphs. We leave the exploration of whether utilizing edge directions and weights can lead to better performance on those of our datasets that have this information for future work.

For experiments with GBDT and tabular deep learning models, we used the source code from the TabR repository (Gorishniy et al., 2024). For experiments with GNNs, we used a modification of the code from the repository of Platonov et al. (2023b). For experiments with BGNN and EBBS, we used the official repositories of these models Ivanov & Prokhorenkova (2021); Chen et al. (2022). Tabular deep learning models are implemented using PyTorch (Paszke et al., 2019), and GNNs are implemented using PyTorch and DGL (Wang et al., 2019).

We train all our GNNs in a full-batch setting, i.e., we do not use any subgraph sampling techniques and train the models on the full graph. Our ResNet baseline is implemented in the same codebase as our GNNs and is thus also trained in the full-batch setting. In contrast, the tabular neural models MLP-PLR and TabR-PLR are trained on random batches of data samples.

Since GBDT and tabular deep learning models are relatively fast, we conducted an extensive hyperparameter search on the validation set — 70 iterations of Bayesian optimization using Optuna (Akiba et al., 2019). Each method was trained until convergence, which is determined after 16 epochs without improvement on the validation set for neural models and 200 iterations for GBDT models. The batch size for neural models was set to 256 when training. In Tables 3 and 4, we provide the hyperparameter search space for tabular models: XGBoost, LightGBM, CatBoost, MLP-PLR, and TabR-PLR.

As GNNs are relatively slower, we ran a less extensive hyperparameter search for them. Specifically, we ran grid search selecting the learning rate from $\{3 \times 10^{-5}, 3 \times 10^{-4}, 3 \times 10^{-3}\}$ and dropout probability from $\{0, 0.2\}$ (note that the highest learning rate of $3 \times 10^{-3}$ often resulted in NaN issues, however, we still included it in our hyperparameter search, as in our preliminary experiments we found it to be beneficial for some of our dataset + model combinations). In our preliminary experiments we found that the performance of our GNNs is quite stable for a wide variety of reasonable architecture hyperparameters values (we found the use of skip-connections and layer normalization to be important for this stability). Hence, for our final experiments, we kept these hyperparameters fixed. Their values were set as follows: the number of graph neighborhood aggregation blocks was set to 3, the hidden dimension was set to 512 (the only exceptions to these two values were made for our largest datasets

Table 3: The hyperparameter search space for GBDT models.

| XGBoost | | LightGBM | | CatBoost | |
|---|---|---|---|---|---|
| Parameter | Distribution | Parameter | Distribution | Parameter | Distribution |
| colsample_bytree | Uniform[0.5, 1.0] | feature_fraction | Uniform[0.5, 1.0] | bagging_temperature | Uniform[0.0, 1.0] |
| gamma | {0.0, LogUniform[0.001, 100.0]} | lambda_l2 | {0.0, LogUniform[0.1, 10.0]} | depth | UniformInt[3, 14] |
| lambda | {0.0, LogUniform[0.1, 10.0]} | learning_rate | LogUniform[0.001, 1.0] | l2_leaf_reg | Uniform[0.1, 10.0] |
| learning_rate | LogUniform[0.001, 1.0] | num_leaves | UniformInt[4, 768] | leaf_estimation_iterations | Uniform[1, 10] |
| max_depth | UniformInt[3, 14] | min_sum_hessian_in_leaf | LogUniform[0.0001, 100.0] | learning_rate | LogUniform[0.001, 1.0] |
| min_child_weight | LogUniform[0.0001, 100.0] | bagging_fraction | Uniform[0.5, 1.0] | | |
| subsample | Uniform[0.5, 1.0] | | | | |

Table 4: The hyperparameter search space for neural tabular models.

| MLP-PLR | | TabR-PLR | |
|---|---|---|---|
| Parameter | Distribution | Parameter | Distribution |
| num_layers | UniformInt[1, 6] | num_encoder_blocks | UniformInt[0, 1] |
| hidden_size | UniformInt[64, 1024] | num_predictor_blocks | UniformInt[1, 2] |
| dropout_rate | {0.0, Uniform[0.0, 0.5]} | hidden_size | UniformInt[96, 384] |
| learning_rate | LogUniform[3e-5, 1e-3] | context_dropout | Uniform[0.0, 0.6] |
| weight_decay | {0, LogUniform[1e-6, 1e-3]} | dropout_rate | Uniform[0.0, 0.5] |
| plr_num_frequencies | UniformInt[16, 96] | learning_rate | LogUniform[3e-5, 1e-3] |
| plr_frequency_scale | LogUniform[0.001, 100.0] | weight_decay | {0, LogUniform[1e-6, 1e-3]} |
| plr_embedding_size | UniformInt[16, 64] | plr_num_frequencies | UniformInt[16, 96] |
| | | plr_frequency_scale | LogUniform[0.001, 100.0] |
| | | plr_embedding_size | UniformInt[16, 64] |

`web-fraud` and `web-traffic`, for which, to avoid OOM issues, we decreased the number of graph neighborhood aggregation blocks to 2, and decreased the hidden dimension to 256). For GNNs with attention-based graph neighborhood aggregation (GAT and GT), the number of attention heads was set to 4. We used the Adam optimizer (Kingma & Ba, 2014) in all our experiments. We trained each model for 1000 steps and then selected the best step based on the performance on the validation set.

Table 5: The hyperparameter search space for specialized models.

| BGNN | | EBBS | |
|---|---|---|---|
| Parameter | Distribution | Parameter | Distribution |
| learning_rate | {0.01, 0.1} | learning_rate | {0.1, 1.0} |
| iter_per_epoch | {10, 20} | propagation_weight | {2.0, 20.0, 50.0} |
| hidden_size | {64, 256} | num_propagation_steps | {2, 5} |
| graph_convolution | {GCN, GAT, AGNN, APPNP} | | |
| use_only_gbdt | {true, false} | | |

When applying deep learning models to tabular data, the preprocessing of numerical features is critically important. In our experiments, we considered two possible numerical feature transformation techniques: standard scaling and quantile transformation to standard normal distribution. We included them in the hyperparameter search for neural models (both tabular ones and GNNs). Note that, when using PLR embeddings for numerical features, we first apply one of the numerical feature transformations discussed above and only then apply PLR embeddings. For categorical features, we used one-hot encoding for all models except for LightGBM and CatBoost, which support the use of categorical features directly and have their specialized strategies for working with them (XGBoost also offers such a feature, but it is currently marked as experimental, and we were not able to make it work). For regression datasets, neural models might perform better if the target variable is transformed. Therefore, in our experiments on regression datasets with neural models (both tabular ones and GNNs), we considered the options of using the original targets or preprocessing targets with standard scaling, including these two options in the hyperparameter search.

Note that PLR embeddings for numerical features have a number of their own hyperparameters: the number of frequencies used, the frequencies scale, and the embedding dimension. For neural tabular models, we simply included these hyperparameters in the Optuna's hyperparameter search space (see Table 4). For GNNs, due to their relatively longer training time, we fixed the number of frequencies to 48 and the embedding dimension to 16 — the default values recommended by the method authors. As for the frequencies scale, which is typically the most important hyperparameter, we searched over

the following set of values: $\{0.01, 0.03, 0.1, 0.3, 1.0, 3.0, 10.0\}$. Specifically, we fixed the values of learning rate, dropout probability, and, for regression experiments, also the regression target transformation, to the best values found in experiments without PLR embeddings, and then searched only over numerical feature transformation (standard scaling or quantile transformation to standard normal distribution) and PLR frequencies scale for our experiments with PLR embeddings. We used the standard original version of PLR numerical feature embeddings (Gorishniy et al., 2022) for all models except for TabR-PLR, for which we used the `lite` version of PLR embeddings in accordance with the official implementation of the model (Gorishniy et al., 2024).

For the specialized methods BGNN and EBBS, we used the experimental pipelines from their official implementations. For BGNN, this pipeline includes hyperparameter selection. Specifically, the hyperparameter tuning is performed over a predefined grid of values. The method uses decision trees of depth 6 and trains for 200 epochs until convergence, which is determined by 20 epochs without improvement on the validation set. As for EBBS, the authors of the method state in their work (Chen et al., 2022) that their method should work universally well across different graph datasets using a default set of hyperparameters. However, we found this not to be the case, and for the sake of completeness performed a moderate hyperparameter search for EBBS around the provided default values. This method also uses decision trees of depth 6 and trains for 2000 epochs, after which the best epoch is selected based on the performance on the validation set. In Table 5, we provide the hyperparameter search space for specialized methods BGNN and EBBS used in our experiments.

# E    COMPUTATION TIME EXAMPLES FOR EXPERIMENTS ON TABGRAPHS DATASETS

In this section, we provide the computation time of our experiments on a subset of dataset + model combinations. The computation cost significantly depends on the dataset and model used. In Table 6 we provide the time required for a single run of 9 models with their optimal hyperparameters: LightGBM (our fastest GBDT model), MLP-PLR (our fastest graph-agnostic neural network model), TabR-PLR (our slowest graph-agnostic neural network model — note that it uses a retrieval mechanism), LightGBM-NFA and MLP-PLR-NFA (to show how neighborhood feature aggregation affects computation time), GraphSAGE (our fastest GNN), GT (our slowest GNN), GraphSAGE-PLR and GT-PLR (to show how PLR embeddings affect computation time). We provide the computation time of these models on 5 datasets: `tolokers-tab` (our smallest dataset), `city-roads-M` (a mid-sized dataset), `city-roads-L` (a dataset that is approximately 2.5 times larger than `city-roads-M`, but otherwise has graph properties very similar to it), `hm-categories` (a dataset that is mid-sized in the number of nodes, but has significantly higher edge density than most datasets from standard graph ML benchmarks), `web-fraud` (our largest dataset).

All the provided experiments were run on an NVIDIA Tesla A100 80GB GPU, except for LightGBM, which was run on AMD EPYC CPUs. Automatic mixed precision was used in all the provided experiments with neural network models except for the GraphSAGE and GraphSAGE-PLR models on `hm-categories` and `web-fraud` datasets, where we encountered `NaN` issues, and thus ran experiments in full precision. We ran multiple experiments for each model + dataset combination to conduct the hyperparameter search (see Appendix H for details) and made 5 runs with different random seeds in each experiment to compute the mean and standard deviation of model performance. Hence, the total amount of runs for each model + dataset combination was quite large.

# F    ISSUES OF PREVIOUSLY USED GRAPH DATASETS WITH TABULAR NODE FEATURES

In this section, we discuss the issues of graph datasets with heterogeneous tabular node features used for model evaluation in the previous studies (Ivanov & Prokhorenkova, 2021; Chen et al., 2022).

First, let us discuss the classification datasets. `dblp` and `slap` datasets are heterogeneous information networks (HINs) that have several relation types, yet only one relation type was used for constructing the graphs. Better results can likely be achieved by modeling these datasets as heterogeneous graphs. Further, these datasets originally have homogeneous features, which were augmented with some graph statistics to make them heterogeneous. `house-class` and `vk-class` datasets

Table 6: Computation time for one model run (with optimal hyperparameters).

| | tolokers-tab | city-roads-M | city-roads-L | hm-categories | web-fraud |
|---|---|---|---|---|---|
| LightGBM | 1s | 15s | 1m 14s | 12s | 6m 26s |
| MLP-PLR | 3s | 2m 29s | 3m 20s | 53s | 1h 17m |
| TabR-PLR | 7s | 3m 9s | 34m | 12m 19s | MLE |
| LightGBM-NFA | 1s | 32s | 1m 55s | 2m 54s | 48m |
| MLP-PLR-NFA | 7s | 1m 2s | 4m 8s | 1m 17s | 4h 10m |
| GraphSAGE | 19s | 33s | 1m 16s | 2m 36s | 35m |
| GT | 54s | 55s | 2m 9s | 11m 53s | 1h 36m |
| GraphSAGE-PLR | 23s | 53s | 2m 6s | 2m 41s | MLE |
| GT-PLR | 58s | 1m 16s | 3m | 11m 56s | MLE |

are originally regression datasets, but they were converted to classification datasets by binning target values, since there was a lack of classification datasets.

Now, let us discuss the regression datasets. First, county and avazu datasets are very small. For our benchmark, we adopt an extended version of avazu dataset, which is significantly larger. For vk dataset, we found that CatBoost, GCN, and GAT achieve values of $R^2$ less than $0.1$ in the user age prediction task used in the previous studies, which shows that the provided node features and graph structure are not very helpful for the task. house dataset originally does not contain a graph at all. For the purpose of applying graph ML methods to it, edges were constructed between properties (nodes) based on their geographic proximity, while the original property features representing geographic coordinates were removed. However, these node features provide no less information than the graph structure (which is based exclusively on them), so we expect that keeping these features and removing the graph will lead to the same or even better predictive performance. Thus, using the graph structure is not necessary for this dataset. The same might also apply to the county dataset, where edges connect counties that share a border, which is strongly related to their geographical position and thus can be encoded using coordinates as additional node features instead of a graph. Note that in our benchmark we have city-roads-M and city-roads-L datasets which include coordinates of the starts and ends of road segments (nodes) in their features. However, in these datasets, edges are based not simply on the physical proximity of road segments, but on whether the road segments are incident to each other and moving from one segment to another is permitted by traffic rules. Note that this information cannot be completely inferred from coordinates alone. We keep coordinates in our data as node features (which are available to all our models) and verify in our experiments that the graph structure still provides benefits to graph-aware models. Generally, we believe that it is important to be very careful if one constructs a graph based on the spatial proximity of nodes for the purpose of applying graph ML methods to the data. It should always be verified that this graph indeed provides benefits to the models beyond what can be achieved by simply using spatial coordinates as node features (which is much easier than adding a graph structure).

## G  LIMITATIONS AND FUTURE WORK

In this work, we exclusively consider the problem of node property prediction in the transductive setting, i.e., when the entire graph, including the test nodes, is available in advance. We choose to focus on it because it is by far the most popular setting in the current graph ML research and captures many real-world applications. However, there are also many applications where the inductive setting, in which test nodes are not available during training, is more realistic. This setting presents new problems such as a lack of historical features for newly added nodes (the cold start problem) and model generalization to a changing graph structure. We believe creating separate benchmarks for this setting is an important direction for future work.

Further, we consider only the most standard setting of homogeneous static graphs, while more complicated settings are possible: heterogeneous graphs, dynamic graphs, graphs with time series as node attributes (spatiotemporal graphs), etc. These settings are also relevant to many real-world applications and can be explored in future works.

