# OpenReview forum: "TabGraphs: A Benchmark and Strong Baselines for Learning on Graphs with Tabular Node Features"
_ICLR.cc/2025/Conference — Submitted to ICLR 2025_

### Official Review · Reviewer_XBq2 · 2024-10-25

**Soundness:** 3
**Presentation:** 3
**Contribution:** 2
**Rating:** 5
**Confidence:** 4

**Summary:**

This paper proposes a benchmark for graph learning on single-table tasks(1 type of node and 1 type of edge), with heterogenous features.
It constructs 11 tasks from 8 different datasets and ran experiments with GBDTs as well as GNNs on the tasks.
The results show that GNNs outperforms GBDTs on the tasks authors proposed.

**Strengths:**

Overall the writing is good and the presentation is clear.
The experiments the authors provide are extensive, including different types of models, tabular dl models, GBDTs, GNNs, etc.

**Weaknesses:**

The novel contribution is limited in this paper.

On page 2, the authors claimed that GNNs are typically evaluated on homogeneous features, but tabular learning typically has heterogenous features. I think they omit the work of PyTorch Frame (Hu, Weihua, et al. "PyTorch Frame: A Modular Framework for Multi-Modal Tabular Learning." arXiv preprint arXiv:2404.00776 (2024). ), which supports encoding heterogenous tabular data through deep encoders and can be used with PyG on downstream GNN tasks. It is good to include comparison to PyTorch Frame + PyG in your paper as it also fits into graph + heterogenous tabular data.

The authors claim on page 4. "In contrast, our work is focused on single-table data with a single type of relationships
between entities in the same table.". However, it seems to me all the graph structure constructed by the authors can be represented by the graph structure in Relbench as well.

I also think the task type is too limited. Currently it only supports node classification and node regression. The authors did not mention link prediction tasks. Link prediction is a common tasks solved by GNNs and is offered by Relbench. It is unfair to say node level task is "the most common setting in modern graph machine learning".

**Questions:**

In the experiment section, the authors only compared the performance scores. I think it's also good to compare the runtime of the methods, as GBDTs can be much slower on larger datasets as compared to GNNs and tabular models.

The authors propose two types of tasks, node regression and classification. I'd like to see some other task types being covered, for example, multi-label classification, or link prediction. For example, in the hnm dataset, can you add a task to predict new items that will be co-purchases?

The authors should make proper comparisons to existing relational learning benchmarks RelBench and 4DBInfer.

Happy to raise the score if the authors can address all the three comments above.

---

> ### Author Response · Authors · 2024-11-21
>
> Thank you for your review! We answer to your comments below.
>
> > On page 2, the authors claimed that GNNs are typically evaluated on homogeneous features, but tabular learning typically has heterogenous features. I think they omit the work of PyTorch Frame (Hu, Weihua, et al. "PyTorch Frame: A Modular Framework for Multi-Modal Tabular Learning." arXiv preprint arXiv:2404.00776 (2024). ), which supports encoding heterogenous tabular data through deep encoders and can be used with PyG on downstream GNN tasks. It is good to include comparison to PyTorch Frame + PyG in your paper as it also fits into graph + heterogenous tabular data.
>
> Thank you for pointing out the PyTorch Frame. This work is indeed very relevant and we will include its discussion in our related work section. However, note that PyTorch Frame and our work are complementary: PyTorch Frame provides a library for embedding tabular data with deep models (with the possibility of applying GNNs from PyG to it, among other things), while we provide a benchmark of tabular datasets with additional graph structure, to which the methods from PyTorch Frame and PyG can be applied. The experiments in the PyTorch Frame paper do not consider such datasets, they only cover standard tabular datasets with no additional graph structure and RDB datasets from RelBench (see our [general response](https://openreview.net/forum?id=a6XE2GJHjk&noteId=BOsaQwgNGg) for a detailed explanation of the differences between our datasets and RDB datasets). We are glad that the intersection of tabular and graph machine learning is starting to attract more attention.
>
> > The authors claim on page 4. "In contrast, our work is focused on single-table data with a single type of relationships between entities in the same table.". However, it seems to me all the graph structure constructed by the authors can be represented by the graph structure in Relbench as well.
>
> Please see our general response for a detailed explanation of the differences between our single-table data setting and the setting of ML for relational databases (RDBs) which is considered in RelBench. While single-table data can be considered as a special case of multi-table RDB data, this is a very important special case with its own specifics and lots of practical applications, and it potentially requires different methods from those used in the general case of ML for RDBs.
>
> > I also think the task type is too limited. Currently it only supports node classification and node regression. The authors did not mention link prediction tasks. Link prediction is a common tasks solved by GNNs and is offered by Relbench. It is unfair to say node level task is "the most common setting in modern graph machine learning".
>
> Link prediction is an important task. However, it provides a lot of degrees of freedom such as negative edge selection for both training and evaluation, which makes designing a benchmark for link prediction a very complicated task. There are works dedicated specifically to this purpose [1]. Thus, we believe that the link prediction task is out of the scope of our work. We would also like to note that most GNN papers only evaluate their models on node-level tasks, so we believe that node level tasks are indeed the most common setting in modern graph ML.
>
> [1] Evaluating Graph Neural Networks for Link Prediction: Current Pitfalls and New Benchmarking (NeurIPS 2023)
>
> > In the experiment section, the authors only compared the performance scores. I think it's also good to compare the runtime of the methods, as GBDTs can be much slower on larger datasets as compared to GNNs and tabular models.
>
> We provide the runtimes of the considered models in Appendix F. Note that graph-agnostic models are typically faster than graph-aware ones since they do not need to process the graph structure (such processing is not very hardware-friendly and thus often takes much time). In particular, due to LightGBM being highly optimized for performance, it is actually the fastest of the methods considered.
>
> > The authors propose two types of tasks, node regression and classification. I'd like to see some other task types being covered, for example, multi-label classification, or link prediction. For example, in the hnm dataset, can you add a task to predict new items that will be co-purchases?
>
> Predicting item co-purchasing is indeed a practically important task. However, we believe it would be more suitable for a work centered on recommender systems. Also see our response about the challenges of designing benchmarks for the link prediction task above.
>
> > The authors should make proper comparisons to existing relational learning benchmarks RelBench and 4DBInfer.
>
> Please see our general response.

---

### Official Review · Reviewer_CzC7 · 2024-11-03

**Soundness:** 2
**Presentation:** 2
**Contribution:** 2
**Rating:** 5
**Confidence:** 5

**Summary:**

The paper explores enhancing predictive accuracy in tabular machine learning by incorporating graph structures, proposing benchmarks to bridge the gap between heterogeneous tabular and homogeneous graph data features. Results indicate that graph neural networks boost the accuracy of tabular predictions, and simple graph-based enhancements can enable traditional models to effectively compete. Valuable insights are provided for both tabular and graph machine learning practitioners.

**Strengths:**

- **Research Topic.** The studied problem is interesting and practically important for tabular machine learning.

- **Experiments.** Experiments are carried out on 10 datasets with 18 baseline methods, which is quite solid. And there are also enough discussion.

**Weaknesses:**

- **Lack of Novelty.** The concept of constructing graph datasets from tabular data is not novel and has been previously suggested in multiple studies.

- **Writing.** The initial paragraph in the 'Machine Learning for Graphs' section of the related works is excessively lengthy, potentially hindering readability. Besides, The main text should provide more details about how tabular data is represented as graph data. The absence of these details in the main text may lead to confusion among readers.

**Questions:**

- More discussion is needed on the differences between this paper and the previous paper on this topic.

- Why there are models that suffer from performance drop after utilizing PLR?

---

> ### Author Response · Authors · 2024-11-21
>
> Thank you for your review! We address your concerns below.
>
> > Lack of Novelty. The concept of constructing graph datasets from tabular data is not novel and has been previously suggested in multiple studies.
>
> While the concept of constructing graph datasets from tabular data is not novel (and we discuss previous works on this topic in our paper), there is a lack of open datasets that represent this type of data (and this lack was acknowledged in previous works), which impedes research on this topic. Thus, we believe constructing a benchmark of such datasets is very important.
>
> > Writing. The initial paragraph in the 'Machine Learning for Graphs' section of the related works is excessively lengthy, potentially hindering readability. Besides, The main text should provide more details about how tabular data is represented as graph data. The absence of these details in the main text may lead to confusion among readers.
>
> We will restructure our related work section to improve its readability. As for how graphs are constructed for tabular data, we specify what relationships are represented by graph edges for each dataset in the main text, and provide details about the graph construction in the Appendix (since these details take up a lot of space, and moving them to the main text will seriously reduce readability and leave almost no space for other parts of our work).
>
> > More discussion is needed on the differences between this paper and the previous paper on this topic.
>
> We discuss the difference between our work and previous works on learning on homogeneous graphs with tabular node features in the “Machine learning for graphs with tabular node features” paragraph of the related work section. In short, previous works focused on building complicated models, while acknowledging the lack of open datasets to evaluate these models. Our work provides such datasets, and also proposes simple modifications to existing tabular and graph ML methods that outperform complex models proposed in previous works. We also discuss the differences between our work and works on ML for relational databases (RDBs) in the last paragraph of the related work section and in Appendix C. In short, ML for RDBs deals with data obtained from multi-table RDBs and represents it as heterogeneous graphs, while our work deals with data obtained from single tables and represents it as homogeneous graphs (see [our general response](https://openreview.net/forum?id=a6XE2GJHjk&noteId=BOsaQwgNGg) for more details).
>
> > Why there are models that suffer from performance drop after utilizing PLR?
>
> There can be many reasons for why PLR does not improve performance across all models and datasets. For instance, PLR introduces additional hyperparameters and trainable parameters — the former can be difficult to choose optimally, while the latter can lead to overfitting.

---

> ### Comment · Reviewer_CzC7 · 2024-12-03
> **Official Comment by Reviewer CzC7**
>
> Thank you very much for your response. I have read it carefully and will maintain my score.

---

### Official Review · Reviewer_SjtV · 2024-11-04

**Soundness:** 1
**Presentation:** 2
**Contribution:** 1
**Rating:** 3
**Confidence:** 5

**Summary:**

The paper introduces a benchmark to evaluate graph ML methods on tabular data with heterogeneous features. The paper hopes to bridge the gap between tabular and graph machine learning studies. It finds that while GNNs can enhance predictive performance, simple adaptations of standard tabular models could also work well.

**Strengths:**

- The experimental results are comprehensive. I appreciate the efforts from the authors to conduct experiments over 10+ baselines and 10+ datasets
- The paper cited a large number of related works, which helped the readers to know more about the field.

**Weaknesses:**

- The problem setting is flawed. I do not agree that graphs with tabular node features are different from relational databases and can be redeemed as a new type of ML problem to work with. Indeed, the authors mentioned that "In contrast, our work is focused on single-table data with a single type of relationships between entities in the same table", and led the readers to Appendix C. But even after carefully reading the Appendix C, the paper still does not explain why the setting is different from a relational database with 2 tables (1 for the original table, the other for the additional relational information within the table). In that case, my understanding is that this paper is only a benchmark for relational databases with 2 tables, and only serves as a special case from the existing relational database benchmarks.
- Moreover, the authors did mention "Very recently, two benchmarks of large-scale relational databases have been proposed: RelBench (Fey et al., 2023) and 4DBInfer (Wang et al., 2024)." in the appendix. Given the high correlation with the proposed domain, they should be highlighted in the main paper and be thoroughly discussed, instead of hiding it within 1 sentence in the appendix. I'm not sure able the reason why the relevant discussions should not be mentioned in the main paper.
- The writing is a bit verbose. For example, the motivation in the introduction is weak. The first 2 paragraphs only introduce why we need tabular learning and graph learning, which is not relevant to the research problem in this paper.
- The paper makes little to none algorithmic contributions. Overall, I think this paper has not met the publication bar for ICLR.

**Questions:**

- Why does this paper not emphasize the high relevance of ML for relational databases? Why should relevant discussions go into the Appendix? The "Machine learning for graphs" paragraph in the related work took almost a page, but it is less relevant to ML for relational databases for this topic since the setting is essentially just a relational database with 2 tables.
- Why working with homogenous graphs a benefit, rather than a limitation? Extending homogeneous GNNs to heterogeneous GNNs only takes a few lines of code in PyG, and it makes the GNN model more expressive. I don't find homogeneous GNNs present benefits with heterogeneous GNNs.

---

> ### Author Response · Authors · 2024-11-21
>
> Thank you for your review! We address your questions and concerns below.
>
> W1. Please see [our general response](https://openreview.net/forum?id=a6XE2GJHjk&noteId=BOsaQwgNGg) for a detailed discussion of the differences between our setting and ML for relational databases (RDBs). Indeed, our single-table data setting can be viewed as a special case of RDBs with one table and additional relational information. However, this is a very important special case with its own specifics and many practical applications, and it potentially requires different methods from those used in the general case of ML for RDBs. In the same way, textual data can be viewed as a special case of data consisting of both texts and images, and appropriate models (vision-language models) and benchmarks for the latter already exist, but it does not mean that the models and benchmarks for purely textual data no longer need to be developed. Both ML for RDBs and our work aim to bring graph ML methods to tabular data, but our benchmark and RDB benchmarks use different sources of data, different ways to represent it, and different methods of working with it. We believe that both directions are important and deserve attention from the community.
>
> W2. Our aim was definitely not to “hide” this discussion. In fact, our discussion of ML for RDBs is rather extensive, which is why most of it was moved to the Appendix. However, we provide a short summary and a reference to the Appendix in the main text. Following your suggestion, we will try to improve the structure of our paper by providing more details in the main text. Further, everything in our discussion of ML for RDBs can be applied to the RelBench and 4DBInfer works (among others mentioned in our paper), thus it is misleading to say that there is only 1 sentence about them.
>
> W3. The aim of our work is to bridge the gap between tabular and graph ML, thus we believe the discussion of these two fields is quite relevant. Nevertheless, we will try to improve the structure of our paper. Also note that already the second paragraph of the introduction discusses examples of realistic scenarios where a graph structure naturally arises in tabular data, which is exactly the focus of our work.
>
> W4. The ICLR 2025 call for papers lists “datasets and benchmarks” as one of the relevant topics for the conference, and we have appropriately specified “datasets and benchmarks” as the primary area of our work. Thus, we believe our work fits the scope of the conference.  We also would like to point out that empirical ML research is impossible without good datasets and benchmarks, so they are of great importance for the research community. Further, one of our contributions is providing simple modifications that help improve performance of existing models and often achieve the strongest results on the newly proposed datasets.
>
> Q1. Please see our comments above and our general response about the differences between our setting and ML for RDBs. We extensively discuss ML for graphs in the related work section because most modern works about graph ML consider homogeneous graphs, which is exactly our setting, so these works are directly relevant to ours. In contrast, ML for RDBs works consider a different setting with heterogeneous graphs obtained from RDB data, thus being less relevant to our work (but still relevant, of course, so we discuss them briefly in the related work section and in much more detail in the Appendix).
>
> Q2. Working with homogeneous graphs is neither a benefit nor a limitation. Some types of data can be represented as homogeneous graphs, and other types of data can be represented as heterogeneous graphs. Appropriate methods need to be used for both types of data, and benchmarks providing such types of data are needed for the development of these methods. There are many GNNs for heterogeneous graphs that are extremely different from GNNs for homogeneous graphs (for example, some of such GNNs are discussed in [1]). Once such models are developed, swapping one model for another indeed takes a few lines of code, but that does not mean that the development and evaluation of these models is not significant. Using a heterogeneous GNN on a homogeneous graph will not provide any additional expressivity, but will only make the model unnecessarily more complex. Further, there is a lot of current research on GNNs for homogeneous graphs (in fact, much more than on GNNs for heterogeneous graphs), which suggests that the community mostly does not consider that GNNs for homogeneous graphs do not have any benefits. While there are of course advantages to trying to cover the most general case possible, such an approach can also unnecessarily complicate the development and application of ML methods. Thus, it is important to be careful about using appropriate methods for different types of data.
>
> [1] Are we really making much progress? Revisiting, benchmarking, and refining heterogeneous graph neural networks (KDD 2021)

---

> > ### Comment · Reviewer_SjtV · 2024-11-27
> > **Thank you for response**
> >
> > Thanks for the response. Now I understand the perspectives of the authors. However, I respectfully disagree that
> > "Note that ML researchers and practitioners generally do not think of single-table data as a special case of RDBs, as evidenced by there being much more research on ML for single-table data (that does not mention RDBs at all) than on ML for RDBs"
> >
> > The claim has factual errors. Many ML researchers are working on RDBs, and I believe most researchers agree that single tables are a special case of RDBs (please point out evidence if a researcher claimed that tabular data is not a special case of RDB). The reason that there is more research on single-table data is more than on RDBs is quite evident: (1) graph deep learning is a relatively new domain, compared to tabular ML that has been around for decades; (2) There were many practical limitations in lack of benchmarks, inefficient implementations, etc, when applying deep learning to RDBs, but today, those limitations have been resolved.
> >
> > Note that I do not try to be picky on a certain sentence in response - the issues lie in the fundamental assumption of the paper. If, as I have pointed out and now the authors also agreed, that "Indeed, our single-table data setting can be viewed as a special case of RDBs"; then, there must be a strong motivation on why single-table data + graph, which is a RDB with 2 tables, are truly special, so that we can publish 1 ICLR papers, or even more high-quality papers, on this topics. Unfortunately, I couldn't find that strong motivation. While other reviewers did not directly comment on this major issue, I can also find their lack of excitement about this work, as evidenced by their scores and comments. Therefore, I would like to maintain my current evaluation.

---

> ### Author Response · Authors · 2024-12-02
>
> Thank you for your reply. Please note that we never claimed that tabular data cannot be viewed as a special case of RDBs (and we are not aware of any researchers claiming that, which is not surprising, because such a claim is incorrect). What we claim is that most researchers in the field of ML for tabular data do not consider viewing general tabular data as a special case of RDBs as a useful approach, as evidenced by the vast majority of research on tabular data not mentioning RDBs at all. As a few examples, [1-3] are recent papers conducting evaluation of a large number of tabular ML models on a large number of diverse single-table datasets. Together, they use more than a hundred tabular datasets. RDBs are not mentioned even a single time in these works. If you are aware of research that studies the single-table data setting and discusses it in the context of RDBs, we would be grateful if you could share it with us, because we are not familiar with any such works.
>
> We would like to emphasize that we do not aim to downplay the importance of ML for RDBs — huge amounts of data are stored in RDBs, and applying appropriate ML methods to them can lead to great benefits. But there are also vast areas of tabular ML research and practice that do not view themselves as a part of the ML for RDBs field, and we aim to show that graph ML methods can be very useful there too.
>
> [1] Why do tree-based models still outperform deep learning on tabular data? (NeurIPS 2022)
>
> [2] When Do Neural Nets Outperform Boosted Trees on Tabular Data? (NeurIPS 2023)
>
> [3] TabReD: Analyzing Pitfalls and Filling the Gaps in Tabular Deep Learning Benchmarks (preprint 2024)

---

### Official Review · Reviewer_hWYk · 2024-11-18

**Soundness:** 2
**Presentation:** 3
**Contribution:** 2
**Rating:** 3
**Confidence:** 3

**Summary:**

In this work, the authors aim to study the usefulness of introducing graphs while working with ML for tabular data. To bridge the gap between graph based ML (E.g. techniques such as GNNs, Deepwalk, etc.) and tabular ML methods (e.g. GBDT) - the authors propose to create a benchmark of graph datasets with heterogenous tabular node features across domains such as fraud detection, road networks, etc - with varying number of nodes, edges and other commonly studied graph based statistics. The authors then use the created benchmark to evaluate different ML methods and their combinations.

**Strengths:**

1. The authors propose multiple datasets which combine graph structure with tabular data -- specifically with heterogenous node data.
2. The authors compare a wide range of methods on the constructed benchmarks

**Weaknesses:**

1. The authors appear to ignore the graph based tabular datasets and tasks given presented concisely in the survey --  https://arxiv.org/abs/2401.02143 which survey prior works which have employed GNNs for tabular data.
2. While the authors state "If it is possible to define meaningful relations between data samples, it is worth trying to convert the given data to a graph and experiment with ML methods that are capable of processing graph information, as it can lead to significant performance gains" - and given its largely a benchmark paper - they do not propose a standardized mechanism or methodology to augment any table with graphs - largely limiting the novelty of the contributions.
4. The analysis accompanying the results presented on the benchmarks also feel handwavy - "it is important to experiment
with different design choices" - seems to suggest - it is prudent for the downstream users to try every possible method and architecture, and therefore not providing a way to prune the search space of methods or architectures for a given dataset/ problem
5. As the paper notes, the graph methods proposed here are not novel and have been employed by works such as https://arxiv.org/abs/2004.05718 and others -- even for heterogenous node data - and therefore the paper lacks novelty from the method perspective as well.

**Questions:**

Please address the weaknesses section as applicable

---

> ### Author Response · Authors · 2024-11-21
>
> Thank you for your review! We address your concerns below.
>
> 1. Please note that this survey does not explicitly discuss any existing datasets. We have checked which datasets are used by other works discussed in the relevant sections of this survey, and these datasets are either not open, or are among the datasets that that have various flaws, which we discuss in Appendix F, or do not have explicit graph structure at all (it is suggested to construct graph edges based on feature similarity or to learn graph edges from data in an end-to-end way, which are not the approaches of our work). Thus, we believe that our benchmark is an important contribution to the field. We provide open realistic datasets with explicit graph structure which is based on additional external information not contained in node features.
>
> 2. Our experience suggests that there can be no “standardized mechanism or methodology to augment any table with graphs”. Data from different domains requires different ways of constructing a meaningful graph, and for some types of data there is probably no way to construct a meaningful graph at all. However, there are many important real-world applications where a graph can be constructed very naturally — social networks, traffic networks, internet, co-purchasing networks, and many other applications, which we discuss in the second paragraph of the introduction. People working with a particular type of data can often very easily come up with a way to construct a meaningful graph from this data. Our datasets provide a wide range of realistic examples of this.
>
> 3. Note that the phrase “it is important to experiment with different design choices” in our work refers specifically to standard GNN architectures, and we have shown that some other more advanced and possibly over-engineered methods do not work as well, thus pruning the search space. Moreover, we believe that fair evaluation of different competing models is a useful result. If there is no one-size-fits-all solution (and our experiments show that this is the case), it is important for researchers and practitioners to know about it so that they can conduct the necessary model selection to obtain better results for their specific data, instead of believing that there is a single model that universally achieves the best results in all situations (which is how many models are presented nowadays — quite often misleadingly).
>
> 4. While techniques similar to NFA have been used in prior works (as we discuss in our paper), our aim was to highlight the usefulness of this method for learning on graphs with tabular node features. This method was previously overlooked in this setting (we are not aware of any prior works that apply NFA-like techniques to tabular node features), whereas it outperforms much more complicated models that have been proposed previously. We believe that showing this is a useful additional contribution of our work. Also note that the paper on PNA that you cite does not experiment with any datasets with tabular node features, and their method of combining multiple neighborhood aggregation is used as part of GNNs, while we propose using multiple neighborhood aggregations as a feature preprocessing step for graph-agnostic models, which is much simpler and allows for using much more computationally efficient models.

---

> > ### Comment · Reviewer_hWYk · 2024-11-25
> > **Thank you for the reply**
> >
> > Thank you for the reply - really appreciate it.
> >
> > Q: Thank you for the pointer to Appendix F. About this line "Generally, we believe that it is important to be very careful if one constructs a graph based on the spatial proximity of nodes for the purpose of applying graph ML methods to the data." - Can you please point to evidence to show that a heuristic based graph construction approach is not ideal for your data (and that only the proposed approach is suitable)

---

> > > ### Author Response · Authors · 2024-12-02
> > >
> > > Thank you for your response and suggestion! Please note that we do not claim that only our proposed approach is suitable for constructing a graph in any of our proposed datasets, and we do not think that it is possible to prove such a statement, since in most cases there is a huge variety of ways to make connections between data samples, and examining each of them in practice is infeasible. However, if we can find _any_ meaningful way to construct a graph that is useful for solving the given prediction task (which is the case for each of our proposed datasets, as can be seen from the comparison of graph-agnostic and graph-aware methods), then we prove that it is at least worth introducing graph modality to the available data and experimenting with graph ML methods.
> > >
> > > Following your suggestion to compare our graph to a heuristic-based graph construction approach, we conducted an additional experiment on the city-roads-M dataset with a different graph. Our original graph for this dataset is constructed based on road connectivity. For this experiment, we instead constructed a heuristic kNN graph based on road coordinates, i.e., a graph that connects each road segment to $k$ closest segments. Specifically, we computed the coordinates of the middle of each road segment and connected road segments using a directed kNN graph with $k = 14$, which is selected in such a way that the resulting graph is at least weakly-connected. Below we provide the values of $R^2$ for GraphSAGE model on the two considered graphs:
> > >
> > > | road connectivity graph | coordinate-based kNN graph |
> > > | --- | --- |
> > > | 73.35 ± 0.58 | 70.16 ± 0.30 |
> > >
> > > As can be seen, the coordinate-based approach to graph construction leads to a significantly lower $R^2$ value. This indicates that the road connectivity network, which is naturally obtained for our dataset, is more useful for the given regression task than the heuristic coordinate-based kNN graph. We do not find it particularly surprising that the kNN graph is not very useful: it is constructed based on road coordinates, which are also available to the model as node features, and thus the model can learn useful information contained in the road coordinates from node features alone without using the graph. In contrast, the road connectivity graph provides additional useful information that is not otherwise available to the model and cannot be easily encoded via node features.

---

### Author Response · Authors · 2024-11-21
**General response**

We thank all the reviewers for their feedback!

Some reviewers asked us to clarify the differences between the setting considered in our work and the setting of ML for relational databases (RDBs) and thus between our benchmark and RDB benchmarks such as RelBench and 4DBInfer. Below we explain these differences in detail.

There are two main axes of difference. The first concerns where the data comes from. In our work, we assume that data comes from a single table but there is additional information available about relations between the rows of this table. Note that single-table data is the standard setting of tabular ML, and such data appears in a vast range of real-world industrial and scientific applications. In contrast, the setting of ML for RDBs assumes that data comes from a relational database, i.e., a collection of tables with additional relationships between entries of different tables. It is true that, as reviewers notice, our single-table data can be represented as an RDB with a single table, and thus can formally be considered a special case of RDB data. However, it is a very important special case with its own specifics and lots of practical applications, so it potentially requires different methods from those used in the general case of ML for RDBs (see the next paragraph). Note that ML researchers and practitioners generally **do not think of single-table data as a special case of RDBs**, as evidenced by there being much more research on ML for single-table data (that does not mention RDBs at all) than on ML for RDBs. And our work aims to highlight that even if tabular data is not stored in an RDB, graph ML methods still can be effectively applied to it.

The second difference concerns how the data is represented after being converted to a graph. Since in our setting the source of data is a single table containing entities of a single type, the resulting graph is homogeneous. In contrast, in ML for RDBs, the source of data is multiple tables, each containing entities of its own type, and thus the resulting graph is heterogeneous, i.e., it has nodes and edges of different types. Different methods have been developed for working with homogeneous and heterogeneous graphs. For instance, there are a lot of specialized GNNs designed specifically for heterogeneous graphs (HAN, GTN, RSHN, HetGNN, MAGNN, HetSANN, HGT, etc.), many of which are very different from standard GNNs that are typically designed with homogeneous graphs in mind (GCN, GraphSAGE, GAT, GIN, etc.). Since the ML methods used for homogeneous and heterogeneous graphs can be very different, we believe that separate benchmarks for graphs of these two types are very useful. While GNNs for heterogeneous graphs can be applied to homogeneous graphs as well, it does not mean that they are the optimal choice for such graphs, so homogeneous graph benchmarks are needed to answer the question of what is the optimal choice, among others.

Finally, we would like to emphasize that due to these differences our work provides a meaningful contribution to both tabular and graph ML fields. In the last paragraph of the introduction section, we explain that our benchmark can serve two main purposes: “First, as discussed above, there are many real-world applications where graph structure can be naturally added to tabular data, and those interested in such applications can test their models on our benchmark. Hence, we hope that the proposed benchmark and new insights obtained using it will lead to a wider adoption of graph machine learning methods to tabular data in industry and science. Second, our benchmark provides an alternative testbed for evaluating GNN performance and, compared to standard graph benchmarks, offers datasets with very different feature types and prediction tasks which are more realistic and meaningful.” Regarding the first purpose, we show that graph ML methods can be effectively used for tabular data that is not stored in RDBs (while RDB benchmarks naturally cannot serve this purpose). As for the second purpose, we provide an alternative benchmark for evaluating GNNs that are not meant to work with heterogeneous graphs (which is the majority of modern GNNs), while RDB benchmarks can only be used to evaluate GNNs designed specifically for heterogeneous graphs.

Overall, our benchmark targets usecases that are significantly different from those of RDB benchmarks. That being said, both ML for RDBs and our work aim to bring graph ML methods to tabular data, which we believe to be potentially a very fruitful direction.

---

### Meta-Review · Area_Chair_BBzn · 2024-12-20

**Metareview:**

While the topic of bridging graph and tabular machine learning is an important and emerging area, the paper under review fails to meet the bar for publication at ICLR due to the following concerns.

Lack of Novelty: The proposed benchmark does not introduce significant methodological or conceptual advancements. The idea of constructing graph datasets from tabular data is well-explored in prior literature. Furthermore, as pointed out by multiple reviewers, the paper's contributions are largely incremental compared to existing benchmarks like RelBench and 4DBInfer.

Problem Setting and Scope: The distinction between single-table datasets with graph structure and relational databases is not convincingly motivated. The responses from the authors fail to justify why the proposed setting warrants an entirely new benchmark, as opposed to being an extension or a special case of relational database benchmarks.

Analysis and Generalization: The work does not provide a standardized mechanism for constructing graphs from tabular data, leaving the benchmark limited to the datasets provided. Additionally, the lack of exploration into broader task types, such as link prediction or multi-label classification, further restricts the generality and impact of the contribution.

Writing and Presentation: The manuscript is verbose in parts, with insufficient emphasis on crucial distinctions between the proposed work and related benchmarks. Important discussions are relegated to the appendix, diminishing their accessibility.

In summary, while the effort to create a comprehensive benchmark is appreciated, the paper lacks sufficient novelty, fails to convincingly establish the uniqueness of its setting, and is limited in scope. These shortcomings outweigh the strengths of the experimental rigor and practical dataset contributions. I recommend rejection.

**Additional Comments On Reviewer Discussion:**

The main point is novelty and relation to existing relational database benchmarks. Three reviewers questioned the novelty of the benchmark, arguing that single-table data with graph structure is a special case of relational databases, making the paper incremental compared to existing benchmarks like RelBench and 4DBInfer. The authors argued that single-table data with graph structure is an important special case with distinct characteristics and practical applications. They emphasized differences in data representation, sources, and methods compared to RDB benchmarks. They also clarified that their datasets focus on homogeneous graphs, unlike the heterogeneous graphs used in RDB benchmarks. However, this point remained unconvincing. The distinction was not strong enough to warrant a separate benchmark, and the work appeared incremental. Also, reviewers found that the discussion of important existing works was little, which is unfair.

---

### Decision · Program_Chairs · 2025-01-22

Reject